# Potent and broad-spectrum anti-*Candida* activity of 6α-(3'-methoxy-4'-hydroxybenzoyl)-lup-20(29)-ene-3-one, a triterpenoid from *Paullinia pinnata*

Benjamin Kingsley Harley[ID][1*], Inemesit Okon Ben[2], Theophilus Duku Asamoah[ID][1], Cedric Dzidzor K. Amengor[3], Nii Korley Kortei[4], Priscilla Eghan[5], Theophilus Christian Fleischer[1]

1 Department of Pharmacognosy and Herbal Medicine, School of Pharmacy, University of Health and Allied Sciences, Ho, Ghana, 2 Department of Pharmacology and Toxicology, School of Pharmacy, University of Health and Allied Sciences, Ho, Ghana, 3 Department of Pharmaceutical Chemistry, School of Pharmacy, University of Health and Allied Sciences, Ho, Ghana, 4 Department of Sports Nutrition, School of Sports and Exercise Medicine, University of Health and Allied Sciences, Ho, Ghana, 5 Department of Midwifery, School of Nursing and Midwifery, University of Health and Allied Sciences, Ho, Ghana

* bkharley@uhas.edu.gh

## Abstract

Vulvovaginal candidiasis is a common infection that affects women of reproductive age, with a high prevalence among pregnant women. When left untreated, it can lead to pregnancy complications including miscarriage. The study evaluated the activity of the ethyl acetate (EtOAc) extract of *Paullinia pinnata* and its isolated compound 6α-(3'-methoxy-4'-hydroxybenzoyl)-lup-20(29)-ene-3-one against drug-resistant strains and clinical isolates of *Candida albicans*, *Candida glabrata*, *Candida krusei*, *Candida parapsilosis* and *Candida tropicalis*. Furthermore, the effects of the combinations of 6α-(3'-methoxy-4'-hydroxybenzoyl)-lup-20(29)-ene-3-one with voriconazole, nystatin or caspofungin on the *Candida* strains and isolates as well as its antibiofilm activity against the drug-resistant strains are reported in this study. The EtOAc leaves extract of *P. pinnata* demonstrated considerable activity against the drug-resistant *Candida* strains and clinical isolates with minimum inhibitory concentrations (MICs) of 31.25 to 500 µg/mL when assessed for anti-*Candida* activity using the microbroth dilution method. The extract was column fractionated to obtain five bulk fractions, and from the most active bulk fraction, BF4 (MIC = 3.91–31.25 µg/mL), 6α-(3'-methoxy-4'-hydroxybenzoyl)-lup-20(29)-ene-3-one was isolated. This study is the first reported biological activity of the compound. 6α-(3'-methoxy-4'-hydroxybenzoyl)-lup-20(29)-ene-3-one exhibited strong activity against the panel of drug-resistant *Candida* species with MIC ranging from 0.5 to 16 µg/mL (0.85–27.10 µM). This activity was within similar range to that of ketoconazole (MIC = 1–8 µg/mL) and amphotericin B (MIC = 0.5–2 µg/mL). When combined with voriconazole, nystatin or caspofungin using the checkerboard assay, 6α-(3'-methoxy-4'-hydroxybenzoyl)-lup-20(29)-ene-3-one recorded synergism of 33.33% to 83.33% with the antifungals, with

**Data availability statement:** All relevant data are within the manuscript and its Supporting Information files.

**Funding:** The author(s) received no specific funding for this work.

**Competing interests:** The authors have declared that no competing interests exist.

majority of the synergistic interactions observed against *C. albicans* and *C. glabrata* strains and isolates. 6α-(3'-methoxy-4'-hydroxybenzoyl)-lup-20(29)-ene-3-one also demonstrated anti-biofilm activity. Its $IC_{50}$ values against the formation of biofilms and preformed biofilms of the *Candida* species were 15.30–46.40 µg/mL (25.91–78.60 µM) and 25.40–90.84 µg/mL (43.02–153.86 µM), respectively. In conclusion, *P. pinnata* possesses strong anti-*Candida* activity and its isolated compound, 6α-(3'-methoxy-4'-hydroxybenzoyl)-lup-20(29)-ene-3-one displayed potent anti-*Candida* and anti-biofilm action. These findings identify 6α-(3'-methoxy-4'-hydroxybenzoyl)-lup-20(29)-ene-3-one as a promising anti-*Candida* compound warranting further investigation, including cytotoxicity in mammalian cells, before its therapeutic potential can be fully assessed.

## 1 Introduction

Vulvovaginal candidiasis (VVC) is a common infection among women of child-bearing age caused by *Candida* species, particularly *Candida albicans* [1]. This fungal infection is depicted by vulvovaginal discharge and other inflammatory signs like itch and redness detected in the vulva and vaginal mucosa caused by an over-growth of *Candida species* which usually are present as dormant vaginal commensals [2]. An estimated 70–75% of women get an episode of vulvovaginal candidiasis at least once throughout their life [3], with half of these women experiencing a second episode and 7–10% having recurrent infection [4]. Pregnant women are highly vulnerable to vulvovaginal candidiasis due to the physiological changes that occur during pregnancy such as elevated hormonal levels and increased vaginal glycogen secretion with corresponding acidification of medium that promote *Candida* proliferation [5].

*Candida albicans* remains the main aetiological specie of vulvovaginal candidiasis (85–95% of cases). However, increasing prevalence of *non-albicans Candida* (NAC) species including *Candida glabrata*, *Candida krusei*, *Candida parapsilosis* and *Candida tropicalis* have emerged globally [6–8]. This shift from the predominance of *C. albicans* in vulvovaginal candidiasis to NAC is concerning as these species often demonstrate reduce sensitivity to azole antifungal drugs [9–11]. This increase in drug resistance and limited new antifungals in the development pipeline necessitates the pressing need for the discovery of new antifungal drugs to tackle the expanding scope of *Candida species.*

*Paullinia pinnata* L. (Sapindaceae) commonly referred to as 'Toa-ntini' in Ghana (Akan dialect) is a woody climber used widely in traditional medicine for the treatment of several diseases [12]. Across the African continent, preparations from the leaves and roots are administered for the treatment of gonorrhoea, threatened abortion, mental disorders, erectile dysfunction, skin ulcers, wounds and microbial infections [13–16]. The plant has been shown to possess diverse pharmacological activities including antibacterial [13,17,18], neuropharmacological [19,20], anthelminthic [21,22] anti-inflammatory [23,24] and wound healing [25] effects. Several bioactive triterpenoids [18,26–28] and polyphenolic [13,28–30] compounds have also been isolated from it.

The present study investigated the antifungal activity of 6α-(3'-methoxy-4'-hydroxybenzoyl)-lup-20(29)-ene-3-one (Fig 1) isolated from the leaves of *P. pinnata* against *Candida* pathogens noted to cause vulvovaginal candidiasis. 6α-(3'-methoxy-4'-hydroxybenzoyl)-lup-20(29)-ene-3-one is a lupene-type triterpenoid previously isolated from the roots of the plant but with no reported biological activity [31]. We investigated 6α-(3'-methoxy-4'-hydroxybenzoyl)-lup-20(29)-ene-3-one for growth inhibition against drug-resistant strains and clinical isolates of *C. albicans*, *C. glabrata*, *C. krusei*, *C. parapsilosis* and *C. tropicalis* and assessed its antibiofilm activity against the reference strains. Furthermore, we evaluated combinations of the compound with conventional antifungal drugs.

## 2 Materials and methods

### 2.1 Reagents and chemicals

Chloramphenicol, miconazole, ketoconazole, clotrimazole, amphotericin B, voriconazole, sabouraud Dextrose Agar (SDA), XTT assay reagent, Mueller-Hinton (MH) agar and RPMI 1640 medium were obtained from Thermo Fisher (Oxoid Limited, Hampshire, UK) whereas fluconazole came from Pfizer (Pfizer Inc., New York, NY, United States) and caspofungin from Sigma Aldrich (Sigma Chemical Co., St. Louis, MO, United States). Organic solvents employed in the study were of analytical grade and were bought from BDH laboratory supplies (Merck Ltd, Lutterworth, UK).

### 2.2 General procedures

Normal phase silica gel 60 (70–230 mesh; AppliChem, GmbH, Darmstadt, Germany) or Sephadex LH-20 (25–100 µm; Amersham Biosciences) were used as stationary phases in column chromatography (CC). Pre-coated silica gel 60 plates (0.25 mm thickness) incorporated with fluorescent indicator GF254 were employed in Thin Layer Chromatography

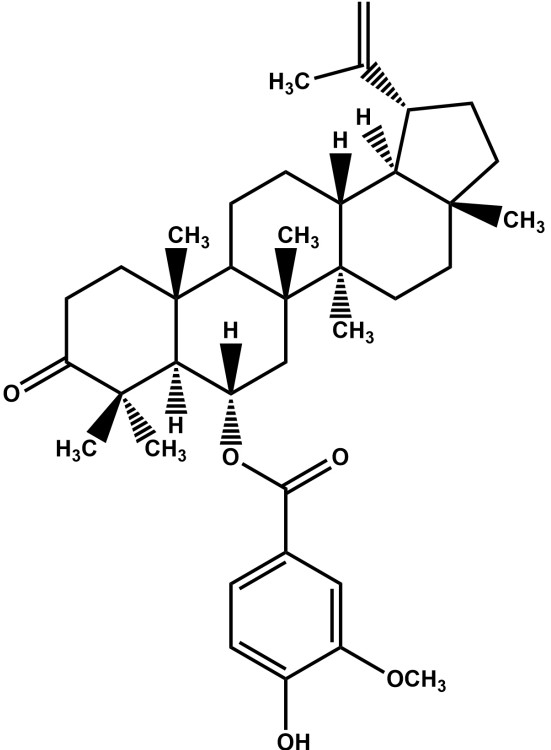

**Fig 1. Chemical structure of 6α-(3'-methoxy-4'-hydroxybenzoyl)-lup-20(29)-ene-3-one.**

(TLC). 1D ($^1$H, $^{13}$C and DEPT135) and 2D (COSY, HSQC and HMBC) NMR spectra were recorded at 25 °C on a Bruker Avance-500 (500 MHz, Bruker BioSpin GmbH, Rheinstetten, Germany), the magnetic field strength (11.7 T). Sample solvent ($CDCl_3$). Chemical shifts (δ) were expressed in parts per million (ppm) using tetramethylsilane (TMS, δ 0.00 ppm) as internal standard and coupling constants (*J*) were measured in Hertz (Hz).

## 2.3 Plant material collection and processing

The leaves of *Paullinia pinnata* were harvested from Sokode, Volta Region in September 2023 with permission of the landowner and identified by Mr. Alfred Agyemang, a medical herbalist at the Institute of Traditional and Alternative Medicine, University of Health and Allied Sciences (UHAS). Voucher specimen was then deposited at the herbarium facility of the of the Department of Pharmacognosy and Herbal Medicine of the same university (Voucher specimen numbers: UHAS/PCOG/2023/L005). The collected leaves were cleaned thoroughly under running water, chopped into pieces and oven dried at 45 °C for 60 hours before grinding into coarse powder.

## 2.4 Extraction and isolation of compound

The powdered leaves of *P. pinnata* (950 g) were extracted by maceration with ethyl acetate (EtOAc, 2.5 L) (3 x 3 days) at room temperature with occasional agitation. Preliminary screening against the *Candida* reference strains was carried out with petroleum ether, ethyl acetate and methanol extracts. The results (data not shown) confirmed greater anti-*Candida* activity in the ethyl acetate extract compared to petroleum ether and methanol extracts

The extract obtained was evaporated to dryness on a rotatory evaporator at 40 °C [32]. The weight of the extracts obtained was 14.05 g (Yield: 1.48%).

A part of the EtOAc extract (12 g) was column chromatographed on silica gel eluting with dichloromethane – methanol ($CH_2Cl_2$: MeOH [v/v, from 90:1–1:1]). TLC profiling of the eluates (100 mL) enabled the bulking of five (5) fractions: BF1 (3.5 g), BF2 (0.44 g), BF3 (0.9 g), BF4 (2.08 g) and F5 (1.2 g). Bulked fraction F4 was subjected to column chromatography on silica gel with petroleum ether – ethyl acetate [Pet ether-EtOAc – 3:7–0:1] and further purification on Sephadex LH-20 with $CH_2Cl_2$: MeOH (1:1) to yield an amorphous powder (1.15 g, 0.12%) [33].

## 2.5 Antifungal assay

### 2.5.1 Fungal strains and growth conditions.
The reference *Candida* strains *C. albicans* ATCC 10231, *C. glabrata* ATCC 2001, *C. krusei* ATCC 6258, *C. parapsilosis* ATCC 22019 and *C. tropicalis* NRRL Y-12968 were purchased from American Type Culture Collection (ATCC®, Manassas, VA, USA).

Forty (40) clinical isolates of *Candida species* were obtained from the Microbiology Laboratory unit of the Ho Teaching Hospital, Ghana. Fungal colonies of the isolates were sub-cultured separately on SDA augmented with chloramphenicol (50 µg/mL) at 37 °C for 48 h to obtain pure *Candida* isolates. Chloramphenicol was added to SDA to suppress bacterial contamination. The pure isolates were then cultured on HiCrome *Candida* Differential Agar (HiMedia Laboratories, India) for 2 days at 35 °C to produce species-specific colours. The colonies produced were identified by colour, appearance and shape (Table 1). The confirmation of the species of the *Candida* isolates were performed using API ID 32C strips (Biomerieux, France) according to standard microbiological methods [34].

The *Candida* strains and isolates were stored in glycerol stocks in −80 °C freezer. Prior to each experiment, *Candida* strains and isolates were retrieved from −80 °C glycerol stocks and sub-cultured on Sabouraud Dextrose Agar (SDA) at 35 °C for 48 h in a humidified chamber before use.

### 2.5.2 Antifungal susceptibility testing.
The sensitivity of the *Candida* isolates to Fluconazole (25 µg), voriconazole (1 µg), miconazole (10 µg), clotrimazole (10 µg), amphotericin B (10 µg) and nystatin (100 units) was evaluated using the disc diffusion method on Mueller-Hinton (MH) agar with slight modifications [35]. Briefly, 5 mL test tubes containing 0.85% sterile saline solution were inoculated with distinct colonies of *Candida* isolates from the SDA plates. They were

**Table 1. Differentiation of *Candida* with HiCrome *Candida* Differential Agar.**

| Species | Description on HiCrome *Candida* Differential Agar |
|---|---|
| *C. albicans* | Light green, glistening, smooth, convex colonies |
| *C. glabrata* | Cream, glistening, convex, smooth colonies |
| *C. tropicalis* | blue to metallic blue, raised colonies |
| *C. krusei* | Purple, fuzzy, dull, flat, irregular colonies |
| *C. parapsilosis* | Cream to pale pink, glistening, smooth, slightly raised colonies |

then emulsified to form a suspension of turbidity equivalent to 0.5 McFarland standard which compared well to 0.5 McF PhoenixSpec Calibrator (Becton, Dickinson and Company, USA). Thereafter, lawns of the media were seeded in three dimensions using sterile swabs dipped in the prepared inoculum. Fluconazole (25 μg), voriconazole (1 μg), miconazole (10 μg), clotrimazole (10 μg), amphotericin B (10 μg) and nystatin (100 units) loaded disks were then aseptically placed on the media lawns and incubated at 37 °C for 48 h. The zone diameters of antifungal disks were measured using a ruler. The zone diameters were interpreted according to Table 2 below [36].

**2.5.3 *In vitro* antifungal activity of *Paullinia pinnata* leaves extract, bulked fractions and compound.** The antifungal activity of the ethyl acetate extract of *P. pinnata* leaves, the bulked fractions from column chromatography and the isolated compound were evaluated using the broth microdilution technique described in document M27 - A3 by the Clinical and Laboratory Standards Institute (CLSI) (Clinical and Laboratory Standards Institute, 2008) with slight modifications [38].

Briefly, test samples were dissolved in 10% dimethyl-sulfoxide (DMSO)/RPMI-1640 medium and a series of two-fold dilutions was carried out in 96-well microplates. Fungal inoculum (*Candida* reference or clinical isolates) was prepared by suspending distinct colonies from SDA plates into 0.85% sterile saline solution and adjusting the turbidity to a 0.5 McFarland standard, verified against a 0.5 McF PhoenixSpec Calibrator (Becton, Dickinson and Company, USA). The inoculum was then further diluted in RPMI-1640 medium to a final working concentration of $2 \times 10^6$ CFU/mL. 50 μL of inoculums of the *Candida species* standardized to $2 \times 10^6$ CFU/mL in RPMI-1640 medium were added to wells of new 96-well microplates. The wells were then treated with 50 μL of extracts/fractions (2000 μg/mL – 1.95 μg/mL), isolated compound (16 μg/mL – 0.125 μg/mL) positive controls (64 μg/mL – 0.125 μg/mL), or an equivalent volume of the DMSO vehicle (1%). The plates were then covered with a sterile plate sealer, gently shaken, and incubated at 37 °C for 72 hrs. The activity, measured in terms of minimum inhibitory concentration (MIC), was determined visually and spectrophotometrically at 490 nm using a microplate reader (Benchmark Microplate reader; Bio-Rad, CA, United States).

Positive controls were voriconazole, ketoconazole and amphotericin B. The negative controls were untreated culture and DMSO vehicle. A media blank was also used to monitor for media contamination. The experiment was carried out in duplicate (two biological replicates) with three technical replicates per treatment and on different days.

**Table 2. Interpretation of Zone diameters [37].**

| Antifungal agents | Zone diameters | | |
|---|---|---|---|
| | Resistant (mm) | Susceptible Dose Dependent (mm) | Susceptible (mm) |
| Fluconazole (25 μg) | ≤ 14 | 15–18 | ≥ 19 |
| Voriconazole (1 μg) | ≤ 13 | 14–16 | ≥ 17 |
| Miconazole (10 μg) | ≤ 11 | 12–19 | ≥ 20 |
| Clotrimazole (10 μg) | ≤ 11 | 12–19 | ≥ 20 |
| Nystatin (100 units) | ≤ 16 | 17–24 | ≥ 25 |
| Amphotericin B (10 μg) | ≤ 9 | 10–14 | ≥ 15 |

The ethyl acetate extract and bulked fractions were considered active when MIC was < 100 µg/mL, moderately active when MIC ranged from 100 to 500 µg/mL, and weakly active when MIC was from 501 to 1000 µg/mL. Above 1000 µg/mL it was deemed inactive [39].

The antifungal activity of the natural product was interpreted as follows: very strong bioactivity < 3.515 µg/mL; strong bioactivity 3.515–25 µg/mL; moderate bioactivity 26–100 µg/mL; weak bioactivity 101–500 µg/mL; very weak bioactivity 500–2000 µg/mL; and no activity above 2000 µg/mL [40].

To evaluate the fungicidal activity, aliquots were taken from the well that corresponded to the MIC and three consecutive wells (two higher and one lower). These were inoculated onto SDA plates and incubated at 37 °C for 24 h. The plates were then analysed for the presence or absence of growth [41].

## 2.6 Checkerboard assay for antifungal combination effects

The effect of the combinations of the compound with voriconazole, nystatin and caspofungin were evaluated using the checkerboard technique from EUCAST-AFST guidelines reference [42] with minor modifications. Serial dilutions (two-fold) of samples (compound and antifungal drugs) were prepared in RPMI 1640 medium in 96-well microtiter plate. 50 µL from each dilution of the extracts were transferred into new 96-well microtiter plates in the vertical direction and same quantities of antifungal drugs were added in horizontal direction to obtain various combinations of the compound with the antifungal drugs. Then, 100 µL of inoculum suspension (0.5 McFarland) of either *Candida* reference strain was added to each well followed by incubation at 37°C for 48 h. MIC values were determined spectrophotometrically at 490 nm using a microplate reader (Benchmark Microplate reader; Bio-Rad, CA, United States). Each experiment was carried out in triplicate (three biological replicates) with three technical replicates per treatment and on different days.

The results were analyzed by determining the Fraction Inhibitory Concentration Indices (FICI) calculated as follows: $FICI = FIC_A + FIC_B$, where $FIC_A = (MIC_{CA} / MIC_A)$ and $FIC_B = (MIC_{CB} / MIC_B)$. $MIC_A$ and $MIC_B$ are the Minimum Inhibitory Concentrations (MIC) of drugs A and B alone; and $MIC_{CA}$ and $MIC_{CB}$ are the concentration of drugs A and B at the iso-effective combinations.

The FIC Indices were interpreted as follows: Synergism: FICI ≤ 0.5, Indifference: > 0.5–4.0 and Antagonism: > 4.0 [43].

## 2.7 Antibiofilm action of the isolated compound against the *Candida* strains and isolates

The effects of the isolated compound on the biofilms of the *Candida* species were evaluated under two conditions using the fungal biofilm formation and susceptibility testing methods: inhibition of biofilm formation and disruption of mature biofilms.

**2.7.1 Biofilm formation inhibitory assay.** The inhibitory effects of the compound on the formation of biofilms by the reference *Candida* strains were determined using a previously described method [44]. In brief, 50 µL of RPMI 1640 was added to each well in a flat-bottom 96-well microplate together with 50 µL of the compound in column 1. Serial dilutions were carried out till 10 to obtain concentrations from 512–2 µg/mL. Then, the wells of the plates from columns 1–11 were inoculated with 50 µL of $2 \times 10^6$ cells/mL fungal suspensions. The plates were then incubated for 24 h at 37 °C. After incubation, the media in each well was carefully removed and the plates washed thrice with 100 µL PBS to remove remnant planktonic cells. Afterwards, 100 µL of XTT/menadione solution was added to each well and the plates incubated at 37°C for 2 h in the dark. 80 µL of the resulting-coloured supernatant from each well was then transferred into new microplates for reading at 490 nm using a microplate reader.

**2.7.2 Disruption of mature biofilms assay.** The inhibitory effect of the compound on mature biofilms of the reference *Candida* strains was determined as follows: 100 µL of fungal suspensions of $1 \times 10^6$ cells/mL in RPMI 1640 of each reference *Candida* strain and the clinical isolates were added to wells of flat-bottom 96-well plates and incubated at 37 °C for 24 h to allow for the formation of biofilms. The media from the wells were aspirated carefully so not to touch the preformed biofilms. The wells were then rinsed (2×) with 100 µL PBS to remove planktonic cells. Two-fold serial dilutions

of the compound were made from 1000 to 1.95 µg/mL in another 96-well plate and transferred to the well plate containing the preformed biofilms. Afterwards, the plates were incubated for 24 h at 37 °C. The media in the wells were then removed carefully and the plate washed twice with 100 µL PBS.

100 µL of XTT/menadione solution was added to each well and the plates incubated at 37 °C for 2 h in the dark. 80 µL of the resulting supernatant from each well was then transferred into a new microplate which was read at 490 nm [45].

Percentage inhibitions in both assays were determined as follows (1):

$$\% \text{ Inhibition } = \frac{\text{Absorbance of control } - \text{ Absorbance of treatment}}{\text{Absorbance of Control}} \times 100 \tag{1}$$

The absorbances were analysed with GraphPad for Windows version 8 (GraphPad Prism Software, San Diego, USA).

The experiment was replicated thrice (three biological replicates) in both assays with three technical replicates per treatment and on different days.

## 2.8 Statistical analysis

Data from all experiments are displayed as mean ± SD. Analyses were carried out using one-way analysis of variance (ANOVA) with GraphPad Prism for Windows version 8 (GraphPad Prism Software, San Diego, USA). For the antifungal activity assays, comparisons between two groups were performed using unpaired t-tests with Welch's correction followed by two-tailed p-value determination. For the antibiofilm assays, percentage inhibition data were plotted as sigmoidal dose-response curves using GraphPad Prism version 8, from which the half maximal inhibitory concentration ($IC_{50}$) values were derived by non-linear regression analysis. For all analysis, $p < 0.05$ was considered statistically significant.

## 2.9 Ethical statement

The study did not involve the use of human subjects, human tissues or organs or animals. The *Candida* clinical isolates used in the study were obtained from the Microbiology Laboratory unit of the Ho Teaching Hospital, Ghana. As such, there was no direct interactions with patients. Ethical approval was not required for the study under the institutional guidelines. The study used existing clinical isolates already collected during routine hospital diagnostic procedures. No identifiable patient data was used. No patient recruitment or intervention took place. No collection permits from the Ghana Forestry Commission were required for plant sample collection as *Paullinia pinnata* is not a protected or endangered species in Ghana.

# 3 Results

## 3.1 Identification of the isolated compound from the leaves of *P. pinnata*

The chemical investigation of the EtOAc leaf extract of *P. pinnata* produced a previously isolated new lupene-type triterpenoid [31]. Its NMR spectra compared favourably with the reported literature and was identified as 6α-(3'-methoxy-4'-hydroxybenzoyl)-lup-20(29)-ene-3-one (Fig 1). The full NMR spectra (1D and 2D NMR) of the compound is outlined in the supplementary data (S1–S6 Figs).

The key $^1$H and $^{13}$C NMR assignment for 6α-(3'-methoxy-4'-hydroxybenzoyl)-lup-20(29)-ene-3-one are presented in Table 3 below.

## 3.2 Microbiological identification of fungi from pregnant women

Table 4 shows that of the forty (40) *Candida spp.* obtained from the Ho Teaching Hospital, Ghana thirty-seven (37) were positive for five strains of *Candida* species, identified as *C. albicans* (n = 11, 29.73%), *C. glabrata* (n = 13, 35.14%), *C. tropicalis* (n = 6, 16.22%), *C. parapsilosis* (n = 2, 5.41%) and *C. krusei* (n = 5, 13.51%).

**Table 3. ¹H, 13C NMR spectra and major HMBC correlations of 6α-(3′-methoxy-4′-hydroxybenzoyl)-lup-20(29)-ene-3-one.**

| C | DEPT/Type | $\delta_C$ | $\delta_H$, multiplicity (*J* in Hz) | Key HMBC Correlations |
|---|---|---|---|---|
| 1 | CH₂ | 39.8 | 1.72 (H-1a) m, 1.91 (H-1b) m | C2 |
| 2 | CH₂ | 33.0 | 2.72 (H-2a) ddd (15.5, 12.0, 6.8), 2.30 (H-2b) ddd (15.2, 9.9, 3.6) | C1, C3 |
| 3 | C | 218.4 | – | |
| 4 | C | 46.7 | – | |
| 5 | CH | 55.9 | 2.13 d (11.4) | C4, C6, C10, C23, C24, C25 |
| 6 | CH | 72.4 | 5.35 dt (11.1, 4.0) | |
| 7 | CH₂ | 39.9 | 1.54 (H-7a), 1.88 (H-7b) m | C6, C8, C26 |
| 8 | C | 41.5 | – | |
| 9 | CH | 49.0 | 1.52 m | C7, C8 |
| 10 | C | 38.50 | – | |
| 11 | CH₂ | 21.9 | 1.45, 1.48 m | |
| 12 | CH₂ | 25.1 | 1.10, 1.75 m | |
| 13 | CH | 37.8 | 1.69 m | |
| 14 | C | 43.0 | – | |
| 15 | CH₂ | 27.5 | 1.00, 1.70 m | |
| 16 | CH₂ | 35.3 | 1.46, 1.32 m | |
| 17 | C | 43.0 | – | |
| 18 | CH | 48.2 | 1.39 m | C12, C13, C14/C17, C16, C20, C28 |
| 19 | CH | 47.9 | 2.39 dt (11.3, 5.9) | C18, C20 |
| 20 | C | 150.8 | – | |
| 21 | CH₂ | 29.8 | 1.41 m | C17, C19, C20, C22 |
| 22 | CH₂ | 39.5 | 1.68 m | |
| 23 | CH₃ | 19.7 | 1.09 s | C3, C4, C5, C24 |
| 24 | CH₃ | 31.3 | 1.33 s | C3, C4, C5, C23 |
| 25 | CH₃ | 17.6 | 0.85 s | C1, C5, C10 |
| 26 | CH₃ | 16.1 | 1.22 s | C9 |
| 27 | CH₃ | 14.5 | 1.00 s | C15 |
| 28 | CH₃ | 18.0 | 0.79 s | C16, C19 |
| 29 | CH₂ | 109.5 | 4.69 (H-29a) br d (2.1), 4.58 (H-29b) br s | C19, C30 |
| 30 | CH₃ | 19.3 | 1.69 | C19, C20, C29 |
| 1' | C | 122.8 | – | |
| 2' | CH | 111.8 | 7.53 d (1.9) | C4', C6', C7' |
| 3' | C | 146.3 | – | |
| 4' | C | 150.1 | – | |
| 5' | CH | 114.1 | 6.95 d (8.4) | C3', C4', C6' |
| 6' | CH | 124.0 | 7.62 dd (8.4, 1.9) | C2', C4', C7' |
| 7' | C | 165.0 | – | |
| – | OCH₃ | 56.1 | 3.94 s | C3' |
| – | OH | – | 6.01 | C3', C4', C5' |

**Table 4. Identification of *Candida* isolates obtained from the Microbiology Laboratory unit of Ho Teaching Hospital.**

| Sample | Initial culture | HiCrome *Candida* agar | Final Identification |
|---|---|---|---|
| 1 | Fungi positive | light green, smooth | *C. albicans* |
| 2 | Fungi positive | light green, smooth | *C. albicans* |
| 3 | Fungi positive | glistening cream, convex | *C. glabrata* |
| 4 | Fungi negative | – | – |
| 5 | Fungi positive | light green, smooth | *C. albicans* |
| 6 | Fungi positive | blue, raised | *C. tropicalis* |
| 7 | Fungi positive | glistening cream, convex | *C. glabrata* |
| 8 | Fungi positive | glistening cream, convex | *C. glabrata* |
| 9 | Fungi positive | light green, smooth | *C. albicans* |
| 10 | Fungi positive | glistening cream, convex | *C. glabrata* |
| 11 | Fungi positive | light green, smooth | *C. albicans* |
| 12 | Fungi positive | light green, smooth | *C. albicans* |
| 13 | Fungi positive | purple, flat | *C. krusei* |
| 14 | Fungi positive | blue, raised | *C. tropicalis* |
| 15 | Fungi positive | glistening cream, convex | *C. glabrata* |
| 16 | Fungi negative | – | – |
| 17 | Fungi positive | purple, flat | *C. krusei* |
| 18 | Fungi positive | pale pink, slightly raised | *C. parapsilosis* |
| 19 | Fungi positive | blue, raised | *C. tropicalis* |
| 20 | Fungi positive | glistening cream, convex | *C. glabrata* |
| 21 | Fungi positive | light green, smooth | *C. albicans* |
| 22 | Fungi positive | glistening cream, convex | *C. glabrata* |
| 23 | Fungi positive | purple, flat | *C. krusei* |
| 24 | Fungi positive | light green, smooth | *C. albicans* |
| 25 | Fungi positive | purple, flat | *C. krusei* |
| 26 | Fungi positive | blue, raised | *C. tropicalis* |
| 27 | Fungi positive | glistening cream, convex | *C. glabrata* |
| 28 | Fungi positive | purple, flat | *C. krusei* |
| 29 | Fungi positive | glistening cream, convex | *C. glabrata* |
| 30 | Fungi positive | blue, raised | *C. tropicalis* |
| 31 | Fungi positive | light green, smooth | *C. albicans* |
| 32 | Fungi positive | glistening cream, convex | *C. glabrata* |
| 33 | Fungi positive | glistening cream, convex | *C. glabrata* |
| 34 | Fungi negative | – | – |
| 35 | Fungi positive | light green, smooth | *C. albicans* |
| 36 | Fungi positive | glistening cream, convex | *C. glabrata* |
| 37 | Fungi positive | light green, smooth | *C. albicans* |
| 38 | Fungi positive | pale pink, slightly raised | *C. parapsilosis* |
| 39 | Fungi positive | blue, raised | *C. tropicalis* |
| 40 | Fungi positive | glistening cream, convex | *C. glabrata* |

### 3.3 Antifungal susceptibility of *Candida* isolates against clinically used antifungal drugs

Generally, the highest resistance rate of the *Candida species* was seen against fluconazole and clotrimazole (62.16%), followed by miconazole (54.05%), voriconazole (43.24%), nystatin (24.32%) and amphotericin B (5.41%). Conversely, the *Candida spp.* was observed to be sensitive to amphotericin B (67.57%), voriconazole (40.54%), nystatin (13.51%), fluconazole and clotrimazole (10.81%), and miconazole (8.11%) in decreasing order. Majority of the *Candida spp.* were also found to be susceptible dose dependent to nystatin (62.16%) (Fig 2).

Apart from amphotericin B, over 50% of *C. albicans* isolates were resistant to all the antifungal drugs. Among the non-*albicans Candida* (NAC), more than 50% were resistant to fluconazole, clotrimazole and miconazole; 76.92% were susceptible dose dependent to nystatin and 69.23% were susceptible to amphotericin B. When treated with voriconazole, 46.15% of the NAC species were susceptible while 34.62% were resistant. Most of the *C. glabrata* and *C. tropicalis* were susceptible dose dependent or resistant to the antifungal drugs except amphotericin B whiles all *C. krusei* were resistant to fluconazole (S1 Table).

### 3.4 Antifungal activities of *P. pinnata* extract

The antifungal activities of the EtOAc extract of *P. pinnata* against the *Candida* reference strains and clinical isolates are displayed in S2 Table. The extract demonstrated considerable antifungal activities against the strains and isolates with minimum inhibitory concentrations (MICs) ranging from 31.25 to 500 µg/mL and minimum fungicidal concentrations (MFCs) of 31.25–2000 µg/mL. Voriconazole, ketoconazole and amphotericin B used as positive controls also showed potent antifungal activities with MICs of 4–32 µg/mL, 1–8 µg/mL and 0.50–2 µg/mL, respectively.

Although there was no significant difference in the activity of the ethyl acetate *P. pinnata* leaves extract against *C. albicans*, *C. glabrata, C. tropicalis* and *C. krusei* strains and isolates, higher concentrations of the extract were required to inhibit the *C. tropicalis* followed by *C. albicans* then *C. glabrata* and *C. krusei* strains and isolates, respectively. However, significantly ($p < 0.05$) higher concentrations of the extract were needed to inhibit the *C. albicans*, *C. glabrata* and *C. tropicalis* strains and isolates than the *C. parapsilosis* strains and isolates (Fig 3).

### 3.5 Antifungal activities of the column fractions of *P. pinnata* ethyl acetate extract

The chromatographic separation and purification of the compound from the EtOAc extract of *P. pinnata* leaves was activity directed by evaluating the antifungal activities of the bulked fractions (BF1 – BF5) against the *Candida* reference strains. Table 5 depicts the antifungal activities of the fractions. They exhibited varying inhibitory actions against the *Candida* strains with bulked fraction F4 demonstrating the highest activity with MIC of 3.91–31.25 µg/mL against the panel of *Candida* strains. The activities of the positive controls were as earlier indicated.

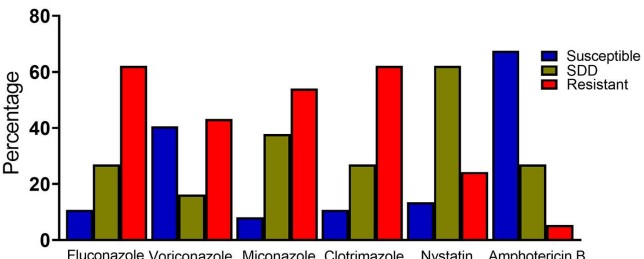

**Fig 2. General *in vitro* susceptibility pattern of the *Candida* isolates to selected antifungal drugs (n = 37).** SDD: Susceptible dose dependent.

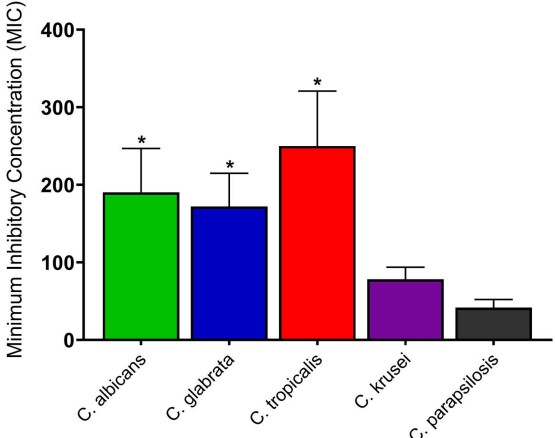

**Fig 3. Antifungal activity of the leaves extract of *P. pinnata* against drug-resistant *Candida* strains and clinical isolates.** Values represent the minimum inhibitory concentration (MIC) expressed in µg/mL and are presented as mean±SD from two independent biological experiments, each performed with three technical replicates per treatment. Comparisons between two groups were carried out using unpaired t-tests with Welch's correction followed by two-tailed *p*-value determination. *$p < 0.05$: comparison between *C. albicans*, *C. glabrata* or *C. tropicalis* and *C. parapsilosis*.

**Table 5. Antifungal activity of the column fractions against *Candida* reference strains.**

| Species/Strain | Minimum Inhibitory Concentration (µg/mL) | | | | | | | |
|---|---|---|---|---|---|---|---|---|
| | BF1 | BF2 | BF3 | BF4 | BF5 | VRC | KTC | AMB |
| *C. albicans* ATCC 10231 | 1000 | 1000 | 2000 | 31.25 | 500 | 8 | 2 | 0.50 |
| *C. glabrata* ATCC 2001 | 500 | 1000 | 1000 | 7.81 | 500 | 4 | 1 | 0.50 |
| *C. tropicalis* NRRL Y-12968 | 1000 | 2000 | 1000 | 15.63 | 500 | 8 | 2 | 1 |
| *C. krusei*, ATCC 6258 | 500 | 1000 | 250 | 7.81 | 250 | 4 | 1 | 0.50 |
| *C. parapsilosis* ATCC 22019 | 250 | 1000 | 250 | 3.91 | 125 | 4 | 1 | 0.50 |

BF1 – BF5: Bulked column fractions of the ethyl acetate extract of *P. pinnata* leaves. VRC: Voriconazole; KTC: Ketoconazole; AMB: Amphotericin B. Experiment was carried out in duplicate with three technical replicates per treatment and on different days.

### 3.6 Antifungal activity of 6α-(3'-methoxy-4'-hydroxybenzoyl)-lup-20(29)-ene-3-one

6α-(3'-methoxy-4'-hydroxybenzoyl)-lup-20(29)-ene-3-one (Fig 1) was investigated for growth inhibition using the broth microdilution technique described by the Clinical and Laboratory Standards Institute (CLSI) with slight modifications against forty-two drug-resistant strains from five (5) species of *Candida* of which five (5) were reference strains and thirty-seven (37) were clinical isolates. The MICs of the compound against the *Candida* strains investigated ranged from 0.5 to 16 µg/mL (0.85–27.10 µM). The compound's MIC was found to range from 1 to 16 µg/mL against *C. albicans*, 0.5 to 4 µg/mL against *C. glabrata*, 1–8 µg/mL against *C. tropicalis*, 0.5 to 2 µg/mL against *C. krusei*, and 0.5 to 1 µg/mL against *C. parapsilosis* (S3 Table).

Although the compound generally exhibited potent activities against the *Candida* species, significantly ($p < 0.05$) higher concentrations of the compound were required to inhibit the *C. albicans* and *C. glabrata* than *C. parapsilosis*. Again, higher concentrations ($p < 0.05$) were needed to inhibit the *C. albicans* compared to *C. krusei* (Fig 4).

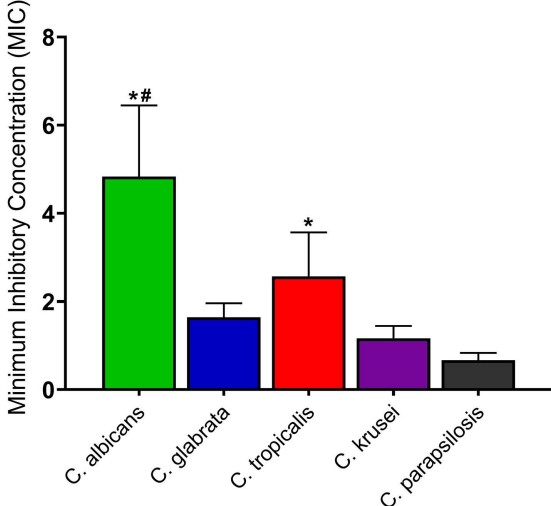

**Fig 4. Antifungal activity of 6α-(3'-methoxy-4'-hydroxybenzoyl)-lup-20(29)-ene-3-one against drug-resistant *Candida* strains and clinical isolates.** Values represent the minimum inhibitory concentration (MIC) expressed in μg/mL and are presented as mean±SD from two independent biological experiments, each performed with three technical replicates per treatment. Comparisons between two groups were carried out using unpaired t-test with Welch's correction followed by two-tailed *p*-value determination. *$p < 0.05$: comparison between *C. albicans* or *C. glabrata* and *C. parapsilosis*. #$p < 0.05$: comparison between *C. albicans* and *C. krusei*.

## 3.7 Antifungal activities of the combinations of 6α-(3'-methoxy-4'-hydroxybenzoyl)-lup-20(29)-ene-3-one with clinically used antifungal agents

The effect of the combinations of 6α-(3'-methoxy-4'-hydroxybenzoyl)-lup-20(29)-ene-3-one with voriconazole, nystatin or caspofungin was investigated using the checkerboard assay. Generally, the MICs of the combinations of the compound with the antifungal drugs against the *Candida* species were either lower or equal to the MICs of the corresponding single agent, and most importantly, none of the combinations exhibited antagonistic interactions (S4 Table). The median fraction inhibitory concentration indices (FICIs) for combinations of the compound with the antifungal drugs were in the range of 0.39–1.30 against the *Candida* species with synergistic action ranging from 33.33% to 83.33%, with 6α-(3'-methoxy-4'-hydroxybenzoyl)-lup-20(29)-ene-3-one – nystatin combination against *C. albicans* being the most prevalent at 83.33% (Table 6). On the other hand, the combination of the compound with nystatin against *C. parapsilosis* did not induce any synergistic action, with FICI range of 0.56–1.13. Around 70% of the detected synergism was against the *C. albicans* and *C. glabrata* strains and isolates. Furthermore, the three remaining *Candida* species demonstrated different patterns of interactions that ranged from synergism to indifference when treated with the combinations of the natural product with the antifungal drugs (Table 6).

## 3.8 Anti-biofilm activity of 6α-(3'-methoxy-4'-hydroxybenzoyl)-lup-20(29)-ene-3-one

### 3.8.1 Biofilm formation inhibition. 6α-(3'-methoxy-4'-hydroxybenzoyl)-lup-20(29)-ene-3-one was investigated for biofilm formation inhibition against the five drug-resistant reference strains. The inhibitory activity, against all *Candida* species tested, measured in terms of half maximal inhibitory concentration (IC$_{50}$) ranged from 15.30–46.40 μg/mL (equivalent to 25.91–78.60 μM). The compound's IC$_{50}$ for *C. albicans* was 46.40±3.64 μg/mL (Fig 5A), for *C. glabrata* was 23.43±1.43 μg/mL (Fig 5B), for *C. tropicalis* was 31.27±2.50 μg/mL (Fig 5C), for *C. krusei* was 15.30±1.39 μg/mL (Fig 5D), and for *C. parapsilosis* was 25.40±1.89 μg/mL (Fig 5E) (Fig 5).

**Table 6. *In vitro* activities of the combinations of 6α-(3'-methoxy-4'-hydroxybenzoyl)-lup-20(29)-ene-3-one with voriconazole, nystatin and caspofungin against the *Candida* species.**

| Combinations/Species | FICI | | % showing the following interactions: | | |
|---|---|---|---|---|---|
| | FICI range | Median FICI | Synergism | Indifference | Antagonism |
| Combination with voriconazole | | | | | |
| *C. albicans* | 0.11–0.75 | 0.39 | 75 (9/12) | 25 (3/12) | 0 (0/12) |
| *C. glabrata* | 0.28–1.24 | 0.46 | 64.29 (9/14) | 35.71 (5/14) | 0 (0/14) |
| *C. tropicalis* | 0.38–1.00 | 0.45 | 57.14 (4/7) | 42.86 (3/7) | 0 (0/7) |
| *C. krusei* | 0.28–0.75 | 0.43 | 66.67 (4/6) | 33.33 (2/6) | 0 (0/6) |
| *C. parapsilosis* | 0.38–1.50 | 1.30 | 33.33 (1/3) | 66.67 (2/3) | 0 (0/3) |
| Combination with nystatin | | | | | |
| *C. albicans* | 0.25–1.02 | 0.42 | 83.33 (10/12) | 16.67 (3/12) | 0 (0/12) |
| *C. glabrata* | 0.19–1.02 | 0.47 | 57.14 (8/14) | 42.86 (6/14) | 0 (0/14) |
| *C. tropicalis* | 0.09–1.00 | 0.75 | 42.86 (3/7) | 57.14 (4/7) | 0 (0/7) |
| *C. krusei* | 0.31–1.24 | 0.68 | 50 (3/6) | 50 (3/6) | 0 (0/6) |
| *C. parapsilosis* | 0.56–1.13 | 1.13 | 0 (0/3) | 100 (3/3) | 0 (0/3) |
| Combination with caspofungin | | | | | |
| *C. albicans* | 0.31–1.25 | 0.47 | 58.33 (7/12) | 41.67 (5/12) | 0 (0/12) |
| *C. glabrata* | 0.11–1.12 | 0.56 | 50 (7/14) | 50 (7/14) | 0 (0/14) |
| *C. tropicalis* | 0.25–0.75 | 0.40 | 57.14 (4/7) | 42.86 (3/7) | 0 (0/7) |
| *C. krusei* | 0.20–1.05 | 0.62 | 50 (3/6) | 50 (3/6) | 0 (0/6) |
| *C. parapsilosis* | 0.25–0.57 | 0.40 | 66.67 (2/3) | 33.33 (1/3) | 0 (0/3) |

FICI: Fraction Inhibitory Concentration Index. S: Synergism for FICI ≤0.5, I: Indifference FICI was > 0.5 to ≤4.0, and A: Antagonism FICI >4.0. Experiment was carried out in triplicate.

**3.8.2 Preformed biofilm inhibition.** 6α-(3'-methoxy-4'-hydroxybenzoyl)-lup-20(29)-ene-3-one was also evaluated for its tendency to disrupt preformed biofilms of the five drug-resistant reference strains. Higher concentrations of the compound were required to inhibit the preformed biofilms compared to prevention of biofilm formation with $IC_{50}$ ranging from 25.40–90.84 μg/mL (equivalent to 43.02–153.86 μM). The $IC_{50}$ of 6α-(3'-methoxy-4'-hydroxybenzoyl)-lup-20(29)-ene-3-one against *C. albicans* was 90.84 ± 8.81 μg/mL (Fig 6A), against *C. glabrata* was 41.99 ± 6.07 μg/mL (Fig 6B), against *C. tropicalis* was 54.82 ± 3.49 μg/mL (Fig 6C), against *C. krusei* was 25.40 ± 4.37 μg/mL (Fig 6D), and against *C. parapsilosis* was 33.97 ± 2.60 μg/mL (Fig 6E).

# 4 Discussion

In our continued search for new antifungal agents, we evaluated the lupene-type triterpenoid, 6α-(3'-methoxy-4'-hydroxybenzoyl)-lup-20(29)-ene-3-one for antifungal activity using *Candida* pathogens circulating in Ghanaian healthcare centres.

Globally, it is estimated that about 35–40% of pregnant women may have vulvovaginal candidiasis with *C. albicans* responsible for more than 90% of all cases, followed by *C. glabrata* [46,47]. This data does no differ significantly from what is recorded in Ghana. A prevalence of 36.7% was reported in the middle belt [48] whiles 30.7% were recorded in the Ho Municipal Hospital [49] and 50.7% in the Ho Teaching Hospital in the Volta Region [5]. *C. albicans* and *C. glabrata* were the most common identified causative agents. Corroborating these statistics, the present study found that of the 40 clinical isolates obtained from the Microbiology Department of Ho Teaching Hospital, 37 were identified as *Candida species* with *C. glabrata* (35.14%) being the most abundant, followed by *C. albicans* (29.73%), *C. tropicalis* (16.22%), *C. krusei* (13.51%) and *C. parapsilosis* (5.41%).

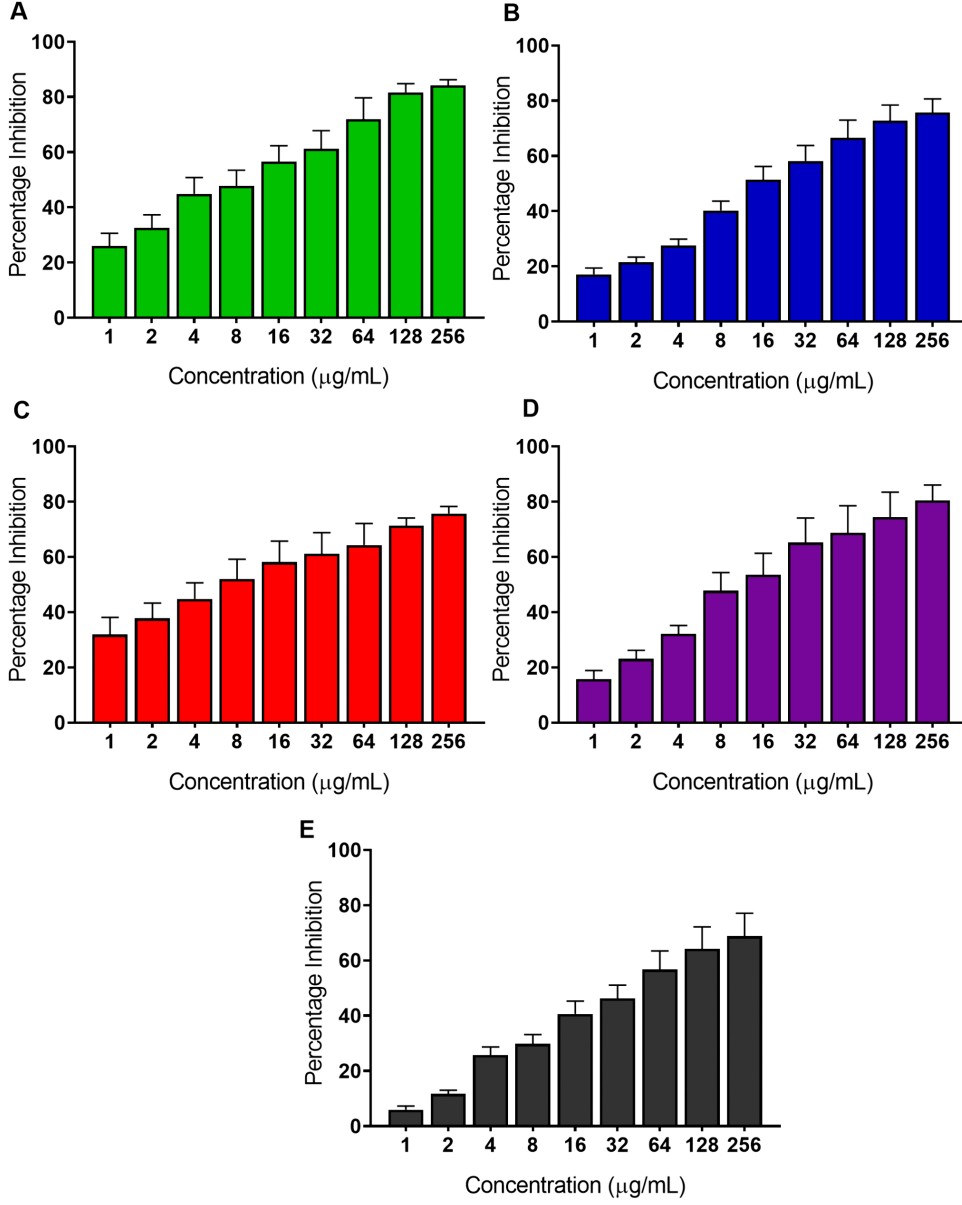

**Fig 5. 6α-(3'-methoxy-4'-hydroxybenzoyl)-lup-20(29)-ene-3-one displays inhibition of biofilm formation against *Candida* species.** Biofilm formation inhibition of the compound against (A) *C. albicans* (B) *C. glabrata* (C) *C. tropicalis* (D) *C. krusei* and (E) *C. parapsilosis.* Data are plotted as mean percentage inhibition ± SD relative to the 1% DMSO vehicle control, from three independent biological experiments. $IC_{50}$ values were determined by non-linear regression analysis of sigmoidal dose-response curves using GraphPad Prism version 8.

Antifungal resistance was also observed in the *Candida* isolates to the clinically used antifungals in Ghana in the study. The *Candida species* demonstrated high resistance to azole antifungals with resistance rate of 62.16% against fluconazole and clotrimazole, 54.05% against miconazole and 43.24% against voriconazole. However, they were less resistant to polyene antifungal agents with resistance rates of 5.41% and 24.32% against amphotericin B and nystatin respectively. The pattern of resistance observed in the azole antifungals, particularly fluconazole is worrying as it remains the most used first-line treatment for vulvovaginal candidiasis [50,51]. The observed resistance to azoles has been attributed to

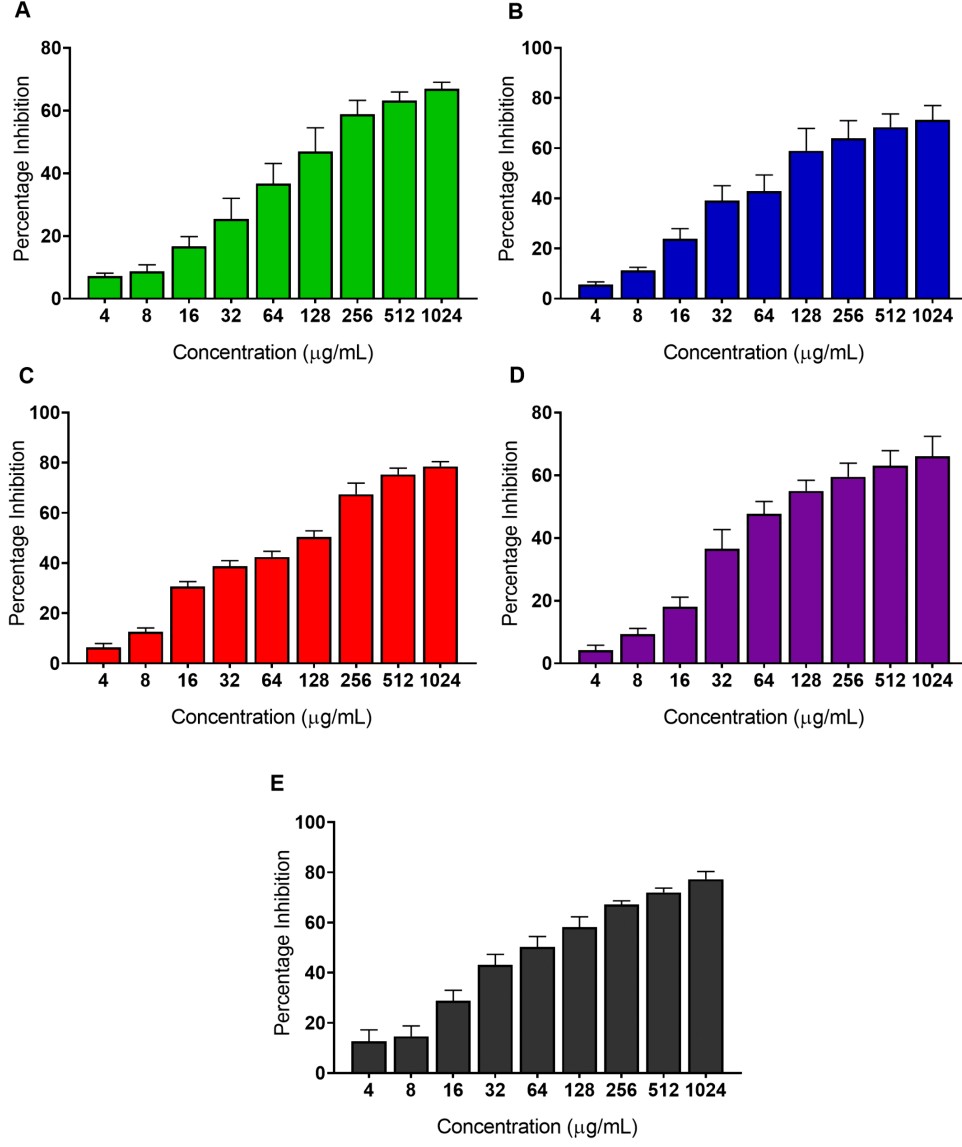

**Fig 6. 6α-(3'-methoxy-4'-hydroxybenzoyl)-lup-20(29)-ene-3-one disrupts preformed biofilms of *Candida* species.** Preformed biofilm inhibition of the compound against (A) *C. albicans* (B) *C. glabrata* (C) *C. tropicalis* (D) *C. krusei* and (E) *C. parapsilosis*. Data are plotted as mean percentage inhibition ± SD relative to the 1% DMSO vehicle control, from three independent biological experiments. IC$_{50}$ values were determined by non-linear regression analysis of sigmoidal dose-response curves using GraphPad Prism version 8.

several factors including mutations in *Candida* ERG11, overexpression of efflux pumps and formation of biofilms [52,53]. Furthermore, the increase in resistance to other antifungal drug classes by *C. albicans* and *non-albicans Candida species* reinforces concerns about the diminishing potency of antifungal therapies and demonstrates the urgent need for the discovery of new antifungal agents capable of overcoming the resistance mechanisms in *Candida* pathogens.

The EtOAc leaves extract of *P. pinnata* exhibited significant antifungal activity against the drug-resistant *Candida* strains and clinical isolates with minimum inhibitory concentrations (MICs) that ranged from 31.25 to 500 μg/mL and minimum fungicidal concentrations (MFCs) of 31.25–2000 μg/mL. Activity-driven chromatographic separation showed

that the anti-*Candida* activity of the EtOAc *P. pinnata* extract was concentrated in bulk column fraction F4 (BF4). While the other column fractions (BF1 – BF3, BF5) exhibited moderate to no antifungal activity against the reference strains of *C. albicans*, *C. glabrata*, *C. krusei*, *C. parapsilosis* and *C. tropicalis* with MIC in the range of 125–2000 µg/mL, BF4 gave MICs of 3.91 µg/mL against *C. parapsilosis* ATCC 22019, 7.81 µg/mL against *C. glabrata* ATCC 2001 and *C. krusei* ATCC 6258, 15.63 µg/mL against *C. tropicalis* NRRL Y-12968 and 31.25 µg/mL against *C. albicans* ATCC 10231 respectively demonstrating strong activity [40]. Further purification of BF4 led to the isolation of 6α-(3'-methoxy-4'-hydroxybenzoyl)-lup-20(29)-ene-3-one. The compound was characterized using NMR analysis and by comparing its spectra data with reported literature [31]. 6α-(3'-methoxy-4'-hydroxybenzoyl)-lup-20(29)-ene-3-one has previously been isolated from the roots of *P. pinnata*. Here, we report for the first time, its isolation from the leaves of the plant.

This study is also the first to investigate the antifungal activity of 6α-(3'-methoxy-4'-hydroxybenzoyl)-lup-20(29)-ene-3-one against multiple *Candida species*, the first reported biological activity of the compound. Our findings show that 6α-(3'-methoxy-4'-hydroxybenzoyl)-lup-20(29)-ene-3-one exerts strong activity against *C. albicans*, *C. glabrata*, *C. tropicalis*, *C. parapsilosis* and *C. krusei* with MIC ranging from 0.5 to 16 µg/mL (0.85–27.10 µM). The MIC values of the compound (0.5–16 µg/mL) fell within similar range to those of ketoconazole (MIC = 1–8 µg/mL) and amphotericin B (MIC = 0.5–2 µg/mL), which were used as control. These comparisons were made under the same standardized *in vitro* microbiological assay conditions. In the study, the *Candida* isolates demonstrated resistance to the azole and polyene antifungals. The potent activity of 6α-(3'-methoxy-4'-hydroxybenzoyl)-lup-20(29)- against drug-resistant *Candida* strains suggests it may overcome existing resistance mechanisms in the *Candida species.* This could reflect a mechanism of action distinct from those of conventional drugs, though this hypothesis requires experimental confirmation.

The antifungal potency demonstrated by 6α-(3'-methoxy-4'-hydroxybenzoyl)-lup-20(29)-ene-3-one in the present study compares favourably with, and in several instances surpasses, that of previously reported antifungal triterpenoids. Among the lupane-type triterpenoids, betulin and betulinic acid are the most extensively studied for antifungal activity. Betulinic acid has been reported to inhibit *C. albicans* with MIC values ranging from 32 to 128 µg/mL [38,54], and betulin has demonstrated MIC values of 64–256 µg/mL against *Candida* species [55], both of which are considerably higher than the MIC values of 1–16 µg/mL recorded for 6α-(3'-methoxy-4'-hydroxybenzoyl)-lup-20(29)-ene-3-one against *C. albicans* in the present study. Among the oleanane-type triterpenoids, oleanolic acid has been reported to exhibit anti-*Candida* activity with MIC values typically in the range of 32–128 µg/mL against *C. albicans* and *C. glabrata*, while its 3-oxo derivative, oleanonic acid, has shown stronger activity in the range of 16–64 µg/mL [56–58]. Ursolic acid, an isomer of oleanolic acid, has similarly demonstrated MIC values of 16–64 µg/mL against *Candida* species [59,60]. Compared with these pentacyclic triterpenoids, the MIC values of 0.5–4 µg/mL recorded for 6α-(3'-methoxy-4'-hydroxybenzoyl)-lup-20(29)-ene-3-one against *C. glabrata* and *C. krusei* represent a substantially higher level of potency. Within the lupane series from *P. pinnata* specifically, other derivatives including the 6β-epimer of the compound [26] and the 3-*O*-isovanilloyl lupene derivative reported by Lasisi et al. [18], have been evaluated for antibacterial activity, with no prior antifungal data reported. The present study therefore establishes 6α-(3'-methoxy-4'-hydroxybenzoyl)-lup-20(29)-ene-3-one as one of the most potent naturally occurring lupane-type triterpenoids reported to date with respect to anti-*Candida* activity to the best of our knowledge. The enhanced potency relative to other triterpenoids may be attributable to the presence of the 6α-vanilloyl ester group, which introduces a polar aromatic substituent onto the lipophilic lupane scaffold. This modification may facilitate interaction with fungal membrane components by combining the membrane-partitioning properties of the triterpenoid core with the hydrogen-bonding capacity of the phenolic hydroxyl and methoxy groups of the vanilloyl moiety This structure-activity relationship hypothesis, however, requires direct experimental validation through systematic modification and testing of structural analogues.

One possible mechanism through which 6α-(3'-methoxy-4'-hydroxybenzoyl)-lup-20(29)-ene-3-one may exert its antifungal activity is through direct disruption of the fungal cell membrane, though this has not been experimentally confirmed. The triterpenoid skeleton is highly lipophilic and has been shown in multiple studies to partition readily into fungal lipid

bilayers [61,62]. The rigid, planar triterpenoid scaffold is capable of intercalating between the acyl chains of membrane phospholipids, thereby altering membrane fluidity, disorganising the bilayer architecture and increasing non-specific membrane permeability [63]. This loss of membrane integrity would compromise essential membrane-dependent processes including nutrient transport, ion homeostasis and maintenance of electrochemical gradients, ultimately leading to cell death. A second possible but as yet confirmed mechanism is direct interaction with membrane ergosterol. As ergosterol is the principal sterol of the fungal cell membrane and plays a critical role in maintaining membrane fluidity, structural organisation and selective permeability, inhibition of its biosynthesis could also be another mechanism that the compound exerts its activity [64].

Some pentacyclic triterpenoids have been shown to inhibit enzymes in the ergosterol biosynthetic pathway, particularly squalene epoxidase and lanosterol synthase, both of which operate upstream of the azole target lanosterol 14α-demethylase [65,66]. Inhibition at these upstream steps would deplete ergosterol from the fungal membrane through a different enzymatic target from azoles, which would account for the potent activity of 6α-(3'-methoxy-4'-hydroxybenzoyl)-lup-20(29)-ene-3-one against azole-resistant strains in which ERG11 is mutated [11], since the compound would not depend on a functional ERG11 target to exert its effect. Furthermore, it is also hypothesized that 6α-(3'-methoxy-4'-hydroxybenzoyl)-lup-20(29)-ene-3-one may disrupt fungal cell wall integrity through inhibition of chitin synthase or β-1,3-glucan synthase [67]. These enzymes are essential for cell wall assembly and maintenance in *Candida* species, and inhibition of either would weaken the cell wall, increase osmotic sensitivity and ultimately cause cell lysis. These possible mechanisms are however hypothetical and require direct experimental validation to confirm.

The formation of biofilms is one of the most important virulence attributes essential for the pathogenicity of several *Candida species* including *C. albicans*, *C. glabrata*, *C. parapsilosis*, *C. tropicalis* and *C. auris* [68,69]. Biofilm formation during infection has been linked to higher mortality rates when compared with infections caused by species incapable of forming biofilms [70]. The development of biofilms functions to counter host immune response by the inhibition of macrophage phagocytosis and antibody activities [68,71]. Furthermore, *Candida* biofilms act as a physical barrier to drug penetration and thereby contribute to their high resistance to antifungal agents [72]. 6α-(3'-methoxy-4'-hydroxybenzoyl)-lup-20(29)-ene-3-one was evaluated for its ability to prevent the formation of biofilms and disrupt fully formed ones in the different reference *Candida* strains. Biofilm formation and preformed *Candida* biofilms were inhibited with half maximal inhibitory concentration ($IC_{50}$) values ranging from 15.30–46.40 µg/mL (equivalent to 25.91–78.60 µM) and 25.40–90.84 µg/mL (equivalent to 43.02–153.86 µM), respectively. These concentrations are significantly higher than those recorded for inhibiting planktonic cells confirming the observation that when *Candida species* exists as biofilms, they acquire additional resistance mechanisms specific and unique to the biofilm state [4,70]. Apart from the presence of the extracellular matrix which acts as a mechanical barrier, the resistance of *Candida* biofilms to antifungals has also been attributed to factors such as upregulation of drug-efflux pumps which are not dependent on antifungal exposure and the presence of persister cells within the biofilm [53,73,74]. The ability of 6α-(3'-methoxy-4'-hydroxybenzoyl)-lup-20(29)-ene-3-one to inhibit both biofilm formation and preformed biofilms is of therapeutic relevance given that biofilm formation is a principal mechanism underlying treatment failure in vulvovaginal candidiasis with conventional antifungal agents demonstrating reduced efficacy against biofilm-associated *Candida* infections [69]. It is hypothesized that 6α-(3'-methoxy-4'-hydroxybenzoyl)-lup-20(29)-ene-3-one may exert its antibiofilm activity by disrupting the composition of extracellular matrix through inhibition of key matrix components, retardation of efflux pump action and inhibiting persister cells. These mechanisms however are hypothetical and further studies including scanning electron microscopy should be carried out to investigate them into details as the antibiofilm activity of the compound is solely based on $IC_{50}$ measurements.

The use of antifungals in combination has emerged as promising therapeutic strategy to overcome resistance in fungal pathogens. Combining antifungal agents, particularly those with different mechanisms of action can increase the potency of the combined molecules resulting in synergy, and extend the spectrum of activity [75,76]. Such combinations also have the potential to reduce dosages thereby lowering toxicity, reducing mortalities and improving the overall treatment outcomes. An example is the combination of amphotericin B with 5-flucytosine used as first-line treatment for

cryptococcal meningitis [77]. The effect of the combination of 6α-(3'-methoxy-4'-hydroxybenzoyl)-lup-20(29)-ene-3-one with voriconazole, nystatin or caspofungin on the *Candida* strains and isolates was investigated by the checkerboard assay. Voriconazole, nystatin and caspofungin were selected as representatives of the three major antifungal classes: azoles, polyenes and echinocandins [78].

Combinations of 6α-(3'-methoxy-4'-hydroxybenzoyl)-lup-20(29)-ene-3-one with the conventional antifungals showed potent synergistic (33.33% to 83.33%) activities across the *Candida* strains and isolates tested. This synergism was particularly evident against isolates displaying azole and polyene resistance, suggesting that such combinations may help overcome established resistance mechanisms in *Candida* pathogens. The high degree of synergism that 6α-(3'-methoxy-4'-hydroxybenzoyl)-lup-20(29)-ene-3-one exhibited in combination with the antifungals also suggest that it may exert its antifungal activity through a pathway completely different from the existing antifungal drugs.

Azoles act by inhibiting lanosterol 14α-demethylase, a key enzyme in the ergosterol biosynthetic pathway [79]. Polyenes exert their effect by forming pores in the cell membrane to facilitate the leakage of intracellular components through binding to ergosterol [80]. Echinocandins on the other hand interfere with fungal cell walls via inhibition of (1,3)-β-D-glucan synthase, an enzyme responsible for the biosynthesis of 1,3-β-D-glucan [81]. The difference in the mechanism of action of 6α-(3'-methoxy-4'-hydroxybenzoyl)-lup-20(29)-ene-3-one to the conventional drug classes might explain the synergistic action of the combinations of 6α-(3'-methoxy-4'-hydroxybenzoyl)-lup-20(29)-ene-3-one with voriconazole, nystatin or caspofungin.

The present study found that the *P. pinnata* leaves possesses antifungal activity against the drug-resistant *Candida species,* and its isolate, 6α-(3'-methoxy-4'-hydroxybenzoyl)-lup-20(29)-ene-3-one has demonstrated strong and broad-spectrum anti-*Candida* activity against standard strains and clinical isolates of *Candida species* implicated in yeast infections. Further studies including evaluating the efficacy and safety of 6α-(3'-methoxy-4'-hydroxybenzoyl)-lup-20(29)-ene-3-one in animal models are therefore warranted to explore this compound as a therapeutic agent for use against *Candida* infections especially in skin and mucosal infections.

A major limitation of the study is that no cytotoxicity evaluation of 6α-(3'-methoxy-4'-hydroxybenzoyl)-lup-20(29)-ene-3-one or the EtOAc extract of *P. pinnata* in mammalian cells was performed. This is a significant limitation because, without cytotoxicity data, it is not possible to determine a therapeutic index and therefore the therapeutic relevance of the antifungal activity observed cannot be fully assessed. Cytotoxicity evaluation of 6α-(3'-methoxy-4'-hydroxybenzoyl)-lup-20(29)-ene-3-one against a panel of mammalian cell lines such as Vero cells and human dermal fibroblasts, given the potential application of the compound in vulvovaginal infections, as well as hepatotoxicity screening using HepG2 cells is therefore identified as the most immediate priority for follow-up investigations.

A second limitation is the absence of scanning electron microscopy (SEM) data to visually confirm the morphological changes in *Candida* biofilms following treatment with 6α-(3'-methoxy-4'-hydroxybenzoyl)-lup-20(29)-ene-3-one. The anti-biofilm activity reported here is based solely on XTT metabolic reduction assays, which measure the metabolic activity of viable cells within biofilms as a proxy for biofilm viability. While this is a well-established and widely used method, it does not provide direct structural evidence of biofilm disruption, extracellular matrix degradation, or changes in *Candida* morphology such as inhibition of hyphal transition in *C. albicans*, which is a critical virulence attribute linked to biofilm formation. SEM studies, complemented by confocal laser scanning microscopy (CLSM) with fluorescent staining of the extracellular matrix, would provide this structural confirmation and are planned as part of subsequent investigations. Future work will therefore address these limitations.

## 5 Conclusion

We report in this study, the anti-*Candida* activity of the ethyl acetate leaves extract of *P. pinnata*. The extract and its isolated compound, 6α-(3'-methoxy-4'-hydroxybenzoyl)-lup-20(29)-ene-3-one have been shown to exert considerable antifungal activity against multiple drug-resistant *Candida species*. 6α-(3'-methoxy-4'-hydroxybenzoyl)-lup-20(29)-ene-3-one

exhibited strong antibiofilm activity by inhibiting the formation of biofilms and preformed biofilms while demonstrating strong synergistic action when combined with voriconazole, nystatin and caspofungin. The findings of the present study are based on *in vitro* experiments. The absence of the cytotoxicity data for 6α-(3'-methoxy-4'-hydroxybenzoyl)-lup-20(29)-ene-3-one is a critical gap that limits the assessment of the therapeutic potential of the compound. Future studies should therefore focus on cytotoxicity evaluation in mammalian cells and subsequent *in vivo* efficacy studies in appropriate animal models in the continued investigation of 6α-(3'-methoxy-4'-hydroxybenzoyl)-lup-20(29)-ene-3-one as a candidate antifungal agent.

## Supporting information

**S1 Table. *In vitro* susceptibility patterns of the clinical strains of *Candida* isolates to fluconazole (25 µg), voriconazole (1 µg), miconazole (10 µg), clotrimazole (10 µg), amphotericin B (10 µg) and nystatin (100 units).** SDD: Susceptible dose-dependent.
(DOCX)

**S2 Table. Antifungal activity of the ethyl acetate extract of the roots of *P. pinnata* against the *Candida* strains and isolates.** Values are expressed in µg/mL. MIC: Minimum inhibitory concentration; MFC: Minimum fungicidal concentration; VRC: Voriconazole; KTC: Ketoconazole; AMB: Amphotericin B; EtOAc: Ethyl acetate. Experiment was carried out in duplicate with three technical replicates per treatment and on different days.
(DOCX)

**S3 Table. Antifungal activity of 6α-(3'-methoxy-4'-hydroxybenzoyl)-lup-20(29)-ene-3-one against the *Candida* strains and isolates.** Values are expressed in µg/mL. MIC: Minimum inhibitory concentration; MFC: Minimum fungicidal concentration; VRC: Voriconazole; KTC: Ketoconazole; AMB: Amphotericin B. Experiment was carried out in duplicate with three technical replicates per treatment and on different days.
(DOCX)

**S4 Table. FIC indices for the combinations of 6α-(3'-methoxy-4'-hydroxybenzoyl)-lup-20(29)-ene-3-one with commonly used antifungal drugs against *Candida* strains and clinical isolates.** INT: Interpretation, FICI: Fraction Inhibitory Concentration Index. S: Synergism for FICI ≤0.5, I: Indifference FICI was > 0.5 to ≤4.0, and A: Antagonism FICI >4.0. Experiment was carried out in triplicate.
(DOCX)

**S1 Fig. $^1$H NMR spectrum of 6α-(3'-methoxy-4'-hydroxybenzoyl)-lup-20(29)-ene-3-one.**
(TIF)

**S2 Fig. $^{13}$C NMR spectrum 6α-(3'-methoxy-4'-hydroxybenzoyl)-lup-20(29)-ene-3-one.**
(TIF)

**S3 Fig. DEPT135 NMR spectrum of 6α-(3'-methoxy-4'-hydroxybenzoyl)-lup-20(29)-ene-3-one.**
(TIF)

**S4 Fig. COSY spectrum of 6α-(3'-methoxy-4'-hydroxybenzoyl)-lup-20(29)-ene-3-one.**
(TIF)

**S5 Fig. HSQC spectrum of 6α-(3'-methoxy-4'-hydroxybenzoyl)-lup-20(29)-ene-3-one.**
(TIF)

**S6 Fig. HMBC spectrum of 6α-(3'-methoxy-4'-hydroxybenzoyl)-lup-20(29)-ene-3-one.**
(TIF)

 

## Acknowledgments

We are grateful to Seeding Labs, Boston, US for the equipment used in the study and to the Microbiology laboratory of the Ho teaching Hospital, Ho for providing the *Candida* clinical isolates.

## Author contributions

**Conceptualization:** Benjamin Kingsley Harley, Theophilus Christian Fleischer.

**Data curation:** Benjamin Kingsley Harley, Theophilus Duku Asamoah, Cedric Dzidzor K. Amengor.

**Formal analysis:** Benjamin Kingsley Harley, Inemesit Okon Ben, Theophilus Duku Asamoah.

**Investigation:** Benjamin Kingsley Harley, Inemesit Okon Ben, Theophilus Duku Asamoah.

**Methodology:** Benjamin Kingsley Harley, Inemesit Okon Ben, Theophilus Duku Asamoah.

**Project administration:** Benjamin Kingsley Harley, Cedric Dzidzor K. Amengor.

**Supervision:** Benjamin Kingsley Harley, Theophilus Christian Fleischer.

**Validation:** Benjamin Kingsley Harley, Inemesit Okon Ben, Theophilus Duku Asamoah.

**Writing – original draft:** Benjamin Kingsley Harley, Inemesit Okon Ben, Nii Korley Kortei, Priscilla Eghan, Theophilus Christian Fleischer.

**Writing – review & editing:** Benjamin Kingsley Harley, Inemesit Okon Ben, Cedric Dzidzor K. Amengor, Nii Korley Kortei, Priscilla Eghan, Theophilus Christian Fleischer.

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
