## [Decision Letter · Decision Letter 0]

26 Jan 2026

PONE-D-25-59885Potent and Broad-spectrum anti-Candida activity of 6α-(3'-methoxy-4'-hydroxybenzoyl)-lup-20(29)-ene-3-one, a triterpenoid from Paullinia pinnataPLOS One

Dear Dr. Harley,

Thank you for submitting your manuscript to PLOS ONE. After careful consideration, we feel that it has merit but does not fully meet PLOS ONE’s publication criteria as it currently stands. Therefore, we invite you to submit a revised version of the manuscript that addresses the points raised during the review process.

**ACADEMIC EDITOR:**

The manuscript addresses an important topic and presents potentially interesting findings. However, based on the reviewers’ evaluations, substantial revisions are required before the manuscript can be considered for publication. The reviewers have raised significant concerns regarding methodological clarity, experimental validation, data interpretation, and overall presentation. In particular, the authors are requested to provide more detailed response to the reviewer's comments and suggestions. Furthermore, the manuscript would benefit from careful revision of the language to improve clarity, consistency, and readability throughout the text.

We look forward to receiving your revised manuscript.

Kind regards,

Vartika Srivastava, PhD

Academic Editor

PLOS One

Journal Requirements:

2. We note that this submission includes NMR spectroscopy data. We would recommend that you include the following information in your methods section or as Supporting Information files:

1) The make/source of the NMR instrument used in your study, as well as the magnetic field strength. For each individual experiment, please also list: the nucleus being measured; the sample concentration; the solvent in which the sample is dissolved and if solvent signal suppression was used; the reference standard and the temperature.

2) A list of the chemical shifts for all compounds characterised by NMR spectroscopy, specifying, where relevant: the chemical shift (δ), the multiplicity and the coupling constants (in Hz), for the appropriate nuclei used for assignment.

3) The full integrated NMR spectrum, clearly labelled with the compound name and chemical structure.

We also strongly encourage authors to provide primary NMR data files, in particular for new compounds which have not been characterised in the existing literature. Authors should provide the acquisition data, FID files and processing parameters for each experiment, clearly labelled with the compound name and identifier, as well as a structure file for each provided dataset. See our list of recommended repositories here: https://journals.plos.org/plosone/s/recommended-repositories

Additional Editor Comments:

The manuscript addresses an important topic and presents potentially interesting findings. However, based on the reviewers’ evaluations, substantial revisions are required before the manuscript can be considered for publication. The reviewers have raised significant concerns regarding methodological clarity, experimental validation, data interpretation, and overall presentation. In particular, the authors are requested to provide more detailed response to the reviewer's comments and suggestions. Furthermore, the manuscript would benefit from careful revision of the language to improve clarity, consistency, and readability throughout the text.

Reviewers' comments:

Reviewer's Responses to Questions

**Comments to the Author**

1. Is the manuscript technically sound, and do the data support the conclusions?

Reviewer #1: Yes

Reviewer #2: Yes

Reviewer #3: Yes

2. Has the statistical analysis been performed appropriately and rigorously? 

Reviewer #1: Yes

Reviewer #2: Yes

Reviewer #3: Yes

3. Have the authors made all data underlying the findings in their manuscript fully available?

Reviewer #1: Yes

Reviewer #2: Yes

Reviewer #3: Yes

4. Is the manuscript presented in an intelligible fashion and written in standard English?

Reviewer #1: Yes

Reviewer #2: Yes

Reviewer #3: Yes

5. Review Comments to the Author

Reviewer #1: The manuscript entitled “Potent and Broad-spectrum anti-Candida activity of 6α-(3′-methoxy-4′-hydroxybenzoyl)-lup-20(29)-ene-3-one, a triterpenoid from Paullinia pinnata” is scientifically sound and well written. The abstract is concise and informative, the methodology is clearly described and technically adequate, and the results support the authors’ conclusions. The study addresses an important area in antifungal drug discovery and contributes valuable data on plant-derived anti-Candida compounds.

The manuscript is generally acceptable for publication; however, the following points require clarification or justification to further strengthen the study.

Specific Comments for Clarification

1. The authors extracted Paullinia pinnata leaves using only ethyl acetate, a moderately polar solvent. Sample extraction with solvents of varying polarity could potentially reveal additional or more potent bioactive compounds. Why were highly polar (e.g., methanol, water) or nonpolar (e.g., hexane) solvents not evaluated? Please justify this choice or acknowledgment of this limitation is recommended.

2. The manuscript does not clearly indicate which solvent was used to resuspend the dried extract prior to bioassays. Please specify the solvent used to resuspend the extract. This information is important for reproducibility and interpretation of antifungal activity.

3. No preliminary phytochemical screening was performed. Such analysis could help correlate observed bioactivity with major classes of secondary metabolites (e.g., terpenoids, flavonoids, phenolics, etc). Please explain why this analysis was not included.

4. While MIC values were determined, MFC testing was not performed by subculturing MIC wells onto Sabouraud Dextrose Agar (SDA). Determining MFC would allow differentiation between fungistatic and fungicidal activity. Please justify its omission or discuss this limitation.

5. The manuscript states that SDA was supplemented with chloramphenicol, but the exact concentration or proportion used is not specified.Please indicate the amount (e.g., µg/mL) and briefly explain why chloramphenicol was included (e.g., suppression of bacterial contamination).

6. The negative control used in the antifungal assays is not clearly defined. If culture medium alone was used, please justify this choice. Ideally, the negative control should include the solvent used to dissolve the extract, to exclude solvent-related effects.

Reviewer #2: In "Potent and Broad-spectrum anti-Candida activity of 6α-(3′-methoxy-4′-hydroxybenzoyl)-lup-20(29)-ene-3-one, a triterpenoid from Paullinia pinnata," a triterpenoid compound's antifungal activity against different Candida species is assessed. The topic is current and relevant, particularly in view of the increasing resistance to antifungals and the ongoing search for novel bioactive natural products.

All things considered, the study answers a significant research question and offers information that PLOS ONE readers might find useful. The experimental work appears generally sound, and the manuscript is clearly written. However, several points should be addressed to improve clarity, rigor, and reproducibility.

The suggested points are intended as minor improvements and do not affect the overall technical soundness of the study or the validity of the authors’ conclusions.

Reviewer #3: Overall Recommendation: Major Revisions

Willingness to review a revised version: Yes.

This study reports the antifungal and antibiofilm properties of Paullinia pinnata extract and its isolated triterpenoid, 6α-(3'-methoxy-4'-hydroxybenzoyl) lup-20 (29)-ene-3-one, against drug-resistant Candida spp. The findings are novel and relevant, particularly amid rising antifungal resistance. However, several issues require clarification and refinement before the manuscript meets PLOS ONE’s standards for methodological transparency and cautious interpretation. The authors evaluated antifungal activity, synergism with standard antifungals, and biofilm inhibition by the isolated triterpenoid from P. pinnata. The compound showed promising MIC values, antibiofilm effects, and synergistic interactions with voriconazole, nystatin, and caspofungin. The work aligns with PLOS ONE’s focus on technical soundness and contributes new biological activity data for a natural compound. However, overstated mechanistic claims, missing safety data, and certain structural issues should be addressed.

Strengths:

• Novel bioactivity report for an isolated triterpenoid

• Use of both clinical isolates and reference strains

• Broad panel of Candida species, including drug-resistant isolates

• Combination and antibiofilm assays add depth

Primary Weaknesses (High Priority):

• No cytotoxicity testing limits translational relevance

• Mechanistic explanations are speculative without supporting experiments

• Overly long discussion sections obscure key findings

• Lack of imaging or molecular analysis for biofilm data

Major Issues

1. Cytotoxicity and Safety Data Are Missing

The manuscript repeatedly mentions therapeutic potential, but there is no safety assessment.

Recommended edit:

“We acknowledge that cytotoxicity studies were not conducted, and future work will assess mammalian cell toxicity to evaluate safety.”

Without safety data, claims of potential therapeutic development should be softened.

2. Mechanistic Claims Are Overstated

Several mechanistic explanations (e.g., efflux pump inhibition, matrix disruption, persister cell effects) are speculative.

Recommended edit:

Replace statements such as:

“The compound could exert its activity by disrupting matrix components…”

with

“These mechanisms are hypothetical and require experimental confirmation.”

3. Biofilm Results Need Clearer Interpretation

No imaging (SEM, confocal) or matrix composition assays were performed. Yet conclusions about structural disruption are made.

Recommended edit:

Emphasize that conclusions are based solely on IC₅₀ measurements.

4. Figures and Tables

The graphs require reformatting the bold text to ensure it remains legible; the bold text is difficult to read in Figures 2-5.

Minor Issues (Improve clarity, but not essential for validity)

• The introduction is too long and repeats epidemiological data.

Suggest reducing the prevalence discussion to ~3–4 sentences.

• Taxonomic formatting:

Ensure Candida genus and species names are italicized consistently.

• Long sentences in discussion:

Several paragraphs contain sentences longer than 40 words; breaking them will improve readability.

• Clarify comparisons to standard antifungals:

The manuscript states that activity is “comparable” to ketoconazole/amphotericin B but does not provide a statistical comparison. Consider rephrasing.

• The abstract is missing a period at the end of the last sentence.

• Lines should be numbered to help editors reference specific text for feedback.

• It should be clarified how the cells were prepared before each experiment. The authors did a great job of showing how they were isolated, sourced, and characterized, but there's a lack of detail on whether fresh liquid cultures were used or whether the plates were used as the primary source of cells.

6. PLOS authors have the option to publish the peer review history of their article (what does this mean?). If published, this will include your full peer review and any attached files.

Reviewer #1: **Yes:** Teshome Geremew

Reviewer #2: **Yes:** Dr. BOKHARI Hassiba I confirm that this review is my own work and that I am not submitting it on behalf of another person.

Reviewer #3: **Yes:** Eduardo Caro

---

## [Author Response · Author response to Decision Letter 1]

26 Feb 2026

RESPONSE TO EDITOR AND REVIWERS’ COMMRENTS: PONE-D-25-59885 - [EMID:94b56f6eecd688ff]

ACADEMIC EDITOR:

The manuscript addresses an important topic and presents potentially interesting findings. However, based on the reviewers’ evaluations, substantial revisions are required before the manuscript can be considered for publication. The reviewers have raised significant concerns regarding methodological clarity, experimental validation, data interpretation, and overall presentation. In particular, the authors are requested to provide more detailed response to the reviewer's comments and suggestions. Furthermore, the manuscript would benefit from careful revision of the language to improve clarity, consistency, and readability throughout the text.

Journal Requirements:

COMMENT 1

RESPONSE

We have reviewed the PLOS ONE style templates and have revised the manuscript to comply with the journal’s formatting requirements, including file naming conventions, heading structure, and reference formatting.

COMMENT 2

We note that this submission includes NMR spectroscopy data. We would recommend that you include the following information in your methods section or as Supporting Information files:

1) The make/source of the NMR instrument used in your study, as well as the magnetic field strength. For each individual experiment, please also list: the nucleus being measured; the sample concentration; the solvent in which the sample is dissolved and if solvent signal suppression was used; the reference standard and the temperature.

2) A list of the chemical shifts for all compounds characterised by NMR spectroscopy, specifying, where relevant: the chemical shift (δ), the multiplicity and the coupling constants (in Hz), for the appropriate nuclei used for assignment.

3) The full integrated NMR spectrum, clearly labelled with the compound name and chemical structure.

We also strongly encourage authors to provide primary NMR data files, in particular for new compounds which have not been characterised in the existing literature. Authors should provide the acquisition data, FID files and processing parameters for each experiment, clearly labelled with the compound name and identifier, as well as a structure file for each provided dataset. See our list of recommended repositories here: https://journals.plos.org/plosone/s/recommended-repositories.

RESPONSE

We appreciate this detailed request. The NMR instrument used in this study was a Bruker Avance-500 (500 MHz), operated at 25°C. This information was already partially stated in Section 2.2 (General Procedures) of the manuscript. We have now expanded the NMR methods description to include all requested details: the nucleus measured (¹H and ¹³C), solvent used (CDCl₃), reference standard (tetramethylsilane, TMS), and temperature. Full integrated NMR spectra are provided as Supporting Information files.

COMMENT 3

In your Methods section, please provide additional information regarding the permits you obtained for the work. Please ensure you have included the full name of the authority that approved the field site access and, if no permits were required, a brief statement explaining why.

RESPONSE

We thank the editor for this reminder. The only field work carried out was the plant collection. The plant material was collected with permission from the local landowner. No collection permits from the Ghana Forestry Commission were required as Paullinia pinnata is not a protected or endangered species in Ghana. We have included that in the ethical statement.

We have also addressed ethical considerations appropriately under the ethical statement section which Reviewer 2 has attested to. The study did not involve the use of human subjects, human tissues or organs or animals. The use of anonymized clinical isolates obtained from a hospital microbiology laboratory is clearly stated and consistent with institutional policies and research integrity standards.

COMMENT 4

Please include captions for your Supporting Information files at the end of your manuscript, and update any in-text citations to match accordingly. Please see our Supporting Information guidelines for more information: http://journals.plos.org/plosone/s/supporting-information.

RESPONSE

We have added captions for all Supporting Information files at the end of the manuscript and have updated all in-text citations to refer to the correctly labelled Supporting Information items.

REVIEWER #1:

The manuscript entitled “Potent and Broad-spectrum anti-Candida activity of 6α-(3′-methoxy-4′-hydroxybenzoyl)-lup-20(29)-ene-3-one, a triterpenoid from Paullinia pinnata” is scientifically sound and well written. The abstract is concise and informative, the methodology is clearly described and technically adequate, and the results support the authors’ conclusions. The study addresses an important area in antifungal drug discovery and contributes valuable data on plant-derived anti-Candida compounds.

The manuscript is generally acceptable for publication; however, the following points require clarification or justification to further strengthen the study.

Specific Comments for Clarification

COMMENT 1

The authors extracted Paullinia pinnata leaves using only ethyl acetate, a moderately polar solvent. Sample extraction with solvents of varying polarity could potentially reveal additional or more potent bioactive compounds. Why were highly polar (e.g., methanol, water) or nonpolar (e.g., hexane) solvents not evaluated? Please justify this choice or acknowledgment of this limitation is recommended.

RESPONSE

We thank the reviewer for this observation. We initially carried out preliminary screening of various solvent extracts (petroleum ether, ethyl acetate and methanol) of the leaves against the reference strains and found that the ethyl acetate extract showed superior anti-Candida activity compared to the petroleum ether and methanol extracts (unpublished data).

We have added the following sentences.

Section 2.4 lines 123 – 126.

Preliminary screening against the Candida reference strains was carried out with petroleum ether, ethyl acetate and methanol extracts. The results (data not shown) confirmed greater anti-Candida activity in the ethyl acetate extract compared to petroleum ether and methanol extracts.

COMMENT 2

The manuscript does not clearly indicate which solvent was used to resuspend the dried extract prior to bioassays. Please specify the solvent used to resuspend the extract. This information is important for reproducibility and interpretation of antifungal activity.

RESPONSE

The dried ethyl acetate extract and isolated compound were dissolved in dimethyl sulfoxide (DMSO) to prepare stock solutions for bioassays. This information was partially described in the biofilm methods section (where DMSO is mentioned) but was not explicitly stated in the antifungal assay methods section in our quest to summarize the methods because we have described them in previous publications. We have expanded the description of the bioassays to ensure reproducibility.

Section 2.5.3

COMMENT 3

No preliminary phytochemical screening was performed. Such analysis could help correlate observed bioactivity with major classes of secondary metabolites (e.g., terpenoids, flavonoids, phenolics, etc). Please explain why this analysis was not included.

RESPONSE

The study was designed as a bioassay-guided fractionation and isolation study focused on the identification of anti-Candida compounds from P. pinnata. Since the primary objective was isolation and characterisation of the bioactive compound a general preliminary phytochemical screening was not central to the study design. A general preliminary phytochemical screening for classes of secondary metabolite is usually not included in publications that report specific isolated bioactive compounds although it is usually carried out as part of the study.

COMMENT 4

While MIC values were determined, MFC testing was not performed by subculturing MIC wells onto Sabouraud Dextrose Agar (SDA). Determining MFC would allow differentiation between fungistatic and fungicidal activity. Please justify its omission or discuss this limitation.

RESPONSE

We determined the minimum fungicidal concentrations (MFC) of the P. pinnata extract and isolated compound as part of the study (Section 2.5.3 lines 194 -196). The results of those determinations are included in the supplementary data (S2 Table and S3 Table). We only employed the MICs of the column fractions to serve as the guide for the isolation (Table 3).

COMMENT 5

The manuscript states that SDA was supplemented with chloramphenicol, but the exact concentration or proportion used is not specified. Please indicate the amount (e.g., µg/mL) and briefly explain why chloramphenicol was included (e.g., suppression of bacterial contamination).

RESPONSE

We thank the reviewer for noting this omission. Chloramphenicol was added to Sabouraud Dextrose Agar at a concentration of 50 µg/mL to suppress bacterial contamination during fungal culture and isolation procedures. This is a standard microbiological practice when working with clinical isolates which may contain mixed flora. We have added this detail to the Methods section.

Section 2.5.1 line 142.

COMMENT 6

The negative control used in the antifungal assays is not clearly defined. If culture medium alone was used, please justify this choice. Ideally, the negative control should include the solvent used to dissolve the extract, to exclude solvent-related effects.

RESPONSE

In a quest to summarise the methods we omitted some important details necessary for reproducibility. We have expanded the methods section particularly section 2.5.3 to include all the negative controls used in the experimentation.

REVIEWER #2:

COMMENT

In "Potent and Broad-spectrum anti-Candida activity of 6α-(3′-methoxy-4′-hydroxybenzoyl)-lup-20(29)-ene-3-one, a triterpenoid from Paullinia pinnata," a triterpenoid compound's antifungal activity against different Candida species is assessed. The topic is current and relevant, particularly in view of the increasing resistance to antifungals and the ongoing search for novel bioactive natural products.

All things considered, the study answers a significant research question and offers information that PLOS ONE readers might find useful. The experimental work appears generally sound, and the manuscript is clearly written. However, several points should be addressed to improve clarity, rigor, and reproducibility.

The suggested points are intended as minor improvements and do not affect the overall technical soundness of the study or the validity of the authors’ conclusions.

RESPONSE

We are grateful to the reviewer for his comments. We acknowledge the minor issues raised and have addressed them to improve clarity, rigor and reproducibility.

REVIEWER #3:

OPENING COMMENTS

This study reports the antifungal and antibiofilm properties of Paullinia pinnata extract and its isolated triterpenoid, 6α-(3'-methoxy-4'-hydroxybenzoyl) lup-20 (29)-ene-3-one, against drug-resistant Candida spp. The findings are novel and relevant, particularly amid rising antifungal resistance. However, several issues require clarification and refinement before the manuscript meets PLOS ONE’s standards for methodological transparency and cautious interpretation. The authors evaluated antifungal activity, synergism with standard antifungals, and biofilm inhibition by the isolated triterpenoid from P. pinnata. The compound showed promising MIC values, antibiofilm effects, and synergistic interactions with voriconazole, nystatin, and caspofungin. The work aligns with PLOS ONE’s focus on technical soundness and contributes new biological activity data for a natural compound. However, overstated mechanistic claims, missing safety data, and certain structural issues should be addressed.

Strengths:

• Novel bioactivity report for an isolated triterpenoid

• Use of both clinical isolates and reference strains

• Broad panel of Candida species, including drug-resistant isolates

• Combination and antibiofilm assays add depth

Primary Weaknesses (High Priority):

• No cytotoxicity testing limits translational relevance

• Mechanistic explanations are speculative without supporting experiments

• Overly long discussion sections obscure key findings

• Lack of imaging or molecular analysis for biofilm data

RESPONSE

We are grateful to the reviewer for the thorough assessment of the work highlighting the strengths and weaknesses of the study. We have addressed the weaknesses to the best of our ability to improve the manuscript.

Major Issues

COMMENT 1

Cytotoxicity and Safety Data Are Missing

The manuscript repeatedly mentions therapeutic potential, but there is no safety assessment.

Recommended edit:

“We acknowledge that cytotoxicity studies were not conducted, and future work will assess mammalian cell toxicity to evaluate safety.”

Without safety data, claims of potential therapeutic development should be softened.

RESPONSE

We agree that the absence of cytotoxicity data is a significant limitation that must be acknowledged. We have softened all claims regarding therapeutic potential throughout the manuscript and have explicitly acknowledged the absence of mammalian cytotoxicity data. We have adopted the recommended language suggested by the reviewer and expanded on this in the Discussion and Conclusion.

We had already acknowledged the absence of the cytotoxicity data as a limitation of the study in the discussion (Section 4 Discussion lines 500 - 502) and had indicated that further work on safety should be done to explore the therapeutic protentional of the compound.

COMMENT 2

Mechanistic Claims Are Overstated

Several mechanistic explanations (e.g., efflux pump inhibition, matrix disruption, persister cell effects) are speculative.

Recommended edit:

Replace statements such as:

“The compound could exert its activity by disrupting matrix components…”

with

“These mechanisms are hypothetical and require experimental confirmation.”

RESPONSE

Indeed, the mechanistic explanations provided in the Discussion were speculative and not by direct experimental evidence in this study. We have revised all such statements throughout the Discussion to make clear that these are hypothetical mechanisms that the compound could have worked by and not conclusions drawn from our data. We have revised those portions of the discussion to reflect them.

Section 4 lines 464 – 467.

COMMENT 3

Biofilm Results Need Clearer Interpretation

No imaging (SEM, confocal) or matrix composition assays were performed. Yet conclusions about structural disruption are made.

Recommended edit:

Emphasize that conclusions are based solely on IC₅₀ measurements.

RESPONSE

We thank the reviewer for this important point. We acknowledge that our biofilm conclusions were solely based on IC₅₀ values determined by the XTT reduction assay, and that no imaging or matrix composition assays were performed. We acknowledge this as a limitation of the study (Section 4 Discussion lines 502 -503) and we have also emphasized that in the discussion and conclusion with the reviewer’s recommendation.

COMMENT 4

Figures and Tables

The graphs require reformatting the bold text to ensure it remains legible; the bold text is difficult to read in Figures 2-5.

RESPONSE

The graphs have been reformatted and all bold text unboldened.

Minor Issues (Improve clarity, but not essential for validity)

ISSUE 1

The introduction is too long and repeats epidemiological data.

Sugge

---

## [Decision Letter · Decision Letter 1]

23 Mar 2026

PONE-D-25-59885R1Potent and Broad-spectrum anti-Candida activity of 6α-(3'-methoxy-4'-hydroxybenzoyl)-lup-20(29)-ene-3-one, a triterpenoid from Paullinia pinnataPLOS One

Dear Dr. Harley,

Thank you for submitting your manuscript to PLOS ONE. After careful consideration, we feel that it has merit but does not fully meet PLOS ONE’s publication criteria as it currently stands. Therefore, we invite you to submit a revised version of the manuscript that addresses the points raised during the review process.

**ACADEMIC EDITOR:**

Thank you for submitting your manuscript entitled “Potent and Broad-spectrum anti-Candida activity of 6α-(3'-methoxy-4'-hydroxybenzoyl)-lup-20(29)-ene-3-one, a triterpenoid from Paullinia pinnata” to our journal. We have now received the reviewers’ comments, which are appended below for your reference.

Based on the reviewers’ evaluations and our editorial assessment, we invite you to revise your manuscript before it can be considered for publication. While the study is of interest and has potential, several important concerns have been raised that need to be addressed to improve the clarity, rigor, and impact of the work.

We look forward to receiving your revised manuscript.

Kind regards,

Vartika Srivastava, PhD

Academic Editor

PLOS One

Journal Requirements:

**Additional Editor Comments:**

Thank you for submitting your manuscript entitled “Potent and Broad-spectrum anti-Candida activity of 6α-(3'-methoxy-4'-hydroxybenzoyl)-lup-20(29)-ene-3-one, a triterpenoid from Paullinia pinnata” to our journal. We have now received the reviewers’ comments, which are appended below for your reference.

Based on the reviewers’ evaluations and our editorial assessment, we invite you to revise your manuscript before it can be considered for publication. While the study is of interest and has potential, several important concerns have been raised that need to be addressed to improve the clarity, rigor, and impact of the work.

Reviewers' comments:

Reviewer's Responses to Questions

**Comments to the Author**

1. If the authors have adequately addressed your comments raised in a previous round of review and you feel that this manuscript is now acceptable for publication, you may indicate that here to bypass the “Comments to the Author” section, enter your conflict of interest statement in the “Confidential to Editor” section, and submit your "Accept" recommendation.

Reviewer #2: All comments have been addressed

Reviewer #3: (No Response)

2. Is the manuscript technically sound, and do the data support the conclusions?

Reviewer #2: Yes

Reviewer #3: Yes

3. Has the statistical analysis been performed appropriately and rigorously? 

Reviewer #2: Yes

Reviewer #3: Yes

4. Have the authors made all data underlying the findings in their manuscript fully available?

Reviewer #2: Yes

Reviewer #3: Yes

5. Is the manuscript presented in an intelligible fashion and written in standard English?

Reviewer #2: Yes

Reviewer #3: Yes

6. Review Comments to the Author

Reviewer #2: A detailed reviewer report is provided as an attachment (uploaded separately), containing a full evaluation, major/minor comments, and a major revision recommendation.

Major strengths: This manuscript presents highly promising preliminary results on the potent and broad-spectrum anti-Candida activity of a triterpenoid [6α-(3'-methoxy-4'-hydroxybenzoyl)-lup-20(29)-ene-3-one] isolated from Paullinia pinnata. The topic is highly relevant given the growing antifungal resistance crisis among Candida species. The study's comprehensive approach—combining antifungal activity, anti-biofilm effects, and synergy with conventional antifungals—is innovative and addresses critical therapeutic gaps. The preliminary data demonstrate substantial potential for plant-derived antifungal development.

Key areas for improvement (detailed in attachment): methodological details (replicates, inoculum preparation), compound characterization (full NMR data/spectra), statistical reporting (tests, p-values, SD), cytotoxicity evaluation, and expanded discussion with mechanism insights.

No concerns regarding dual publication, research ethics, or publication ethics. English requires only minor polishing. Figure legends should be more self-contained. Conclusions appropriately reflect the in vitro scope.

Excellent contribution to open science—worthy of publication following revisions."

Reviewer #3: The authors have made substantial revisions to the original manuscript and have adequately addressed each of the reviewers’ concerns. However, the numbers for the axes in Figures 5 and 6 although improved remain illegible and require correction. Once this issue is resolved, I believe the manuscript will be ready for submission, as significant progress has been made in improving clarity and comprehension, and the work will meet PLOS ONE guidelines.

7. PLOS authors have the option to publish the peer review history of their article (what does this mean?). If published, this will include your full peer review and any attached files.

Reviewer #2: No

Reviewer #3: No

---

## [Author Response · Author response to Decision Letter 2]

24 Apr 2026

RESPONSE TO REVIWERS’ COMMRENTS

REVIEWER 2

Reviewer Report

General evaluation

The manuscript entitled “Potent and Broad-spectrum anti-Candida activity of 6α-(3'-methoxy-4'-hydroxybenzoyl)-lup-20(29)-ene-3-one, a triterpenoid from Paullinia pinnata” investigates the antifungal potential of a plant-derived compound isolated from Paullinia pinnata. The study evaluates antifungal activity against several Candida species, including Candida albicans, and also examines antibiofilm activity and potential synergy with conventional antifungal drugs.

The topic is relevant, as antifungal resistance among Candida species is an important clinical concern. The exploration of bioactive compounds from medicinal plants represents an interesting research direction. The manuscript presents promising preliminary results; however, several aspects of the study require clarification and improvement before the manuscript can be considered for publication.

Major Comments

COMMENT 1: Methodological clarity

The methodology describing the antifungal assays requires more detailed information. The authors should clearly indicate the number of biological and technical replicates, the preparation of fungal inoculum, and the incubation conditions used during the experiments. These details are essential to ensure the reproducibility of the study.

RESPONSE

The number of biological and technical replicates for the antifungal assays have been clearly stated for each experiment performed.

Lines 190 – 191

Lines 212 – 213

Lines 253 - 254

Preparation of the fungal inoculum prior to performing each experiment was stated in section 2.5.1 Lines 148 – 150.

We have also expanded on the inoculum preparation in the in vitro antifungal assay

Section 2.5.3 Lines 175 – 180.

The incubation conditions used during the experiments are also provided for each experiment carried out.

COMMENT 2: Characterization of the isolated compound

Although the compound identification is based on spectroscopic analysis, the manuscript should provide more detailed NMR data. Complete spectroscopic information (chemical shifts, multiplicity, coupling constants) should be clearly presented, and full spectra should be included in the supplementary materials.

RESPONSE

We thank the reviewer for this valid point. NMR spectra (1D: 1H, 13C, DEPT135; 2D: COSY, HSQC, HMBC) were recorded on a Bruker Avance-500 (500 MHz) in CDCl3. The compound was previously characterized by Jackson et al. (2015) [Ref. 31], and our data were in full agreement with the reported literature. We have now added a structured table of key NMR assignments (1H: chemical shifts, multiplicity, coupling constants and 13C shifts) to Section 3.1. Full spectra (S1–S6 Figs.) are included in the supplementary data and are now explicitly cross-referenced in the section.

COMMENT 3: Statistical analysis

The statistical treatment of the experimental data is not sufficiently described. The authors should specify the statistical tests used and present the results with appropriate statistical indicators (e.g., standard deviation, p-values).

RESPONSE

We acknowledge this and have revised the manuscript accordingly. All statistical analyses were performed using GraphPad Prism version 8.0. For two-group comparisons, we used an unpaired t-test with Welch’s correction and two-tailed p-value determination; significance was set at p < 0.05. Results are expressed as mean ± SD throughout. While these details were stated in figure legends (Figs. 3 and 4) and in Section 2.7, they were not described in a dedicated sub-section. We have now added a Statistical Analysis sub-section (Section 2.8) to the Methods and ensured consistent reporting of statistical indicators across all relevant results.

COMMENT 4: Cytotoxicity evaluation

The study reports significant antifungal activity; however, no cytotoxicity evaluation on mammalian cells is presented. This information is important to assess the potential therapeutic relevance of the compound. At minimum, this limitation should be discussed in more detail in the manuscript.

RESPONSE

We fully acknowledge this limitation. Cytotoxicity assays could not be performed in the current study due to the unavailability of appropriate cell culture facilities at our institution at the time. We have now discussed this limitation in substantially greater detail in the Discussion. We emphasize that without a therapeutic index, the translational implications of the antifungal activity cannot yet be fully assessed.

COMMENT 5: Discussion improvement

The discussion section should be expanded to better interpret the results. The authors should provide deeper comparisons with previously reported antifungal triterpenoids and discuss possible mechanisms of action of the compound

RESPONSE

We appreciate this suggestion and have significantly expanded the Discussion. We have added: (1) a comparative analysis positioning the antifungal activity of 6α-(3′-methoxy-4′-hydroxybenzoyl)-lup-20(29)-ene-3-one against other reported antifungal triterpenoids, including betulin, betulinic acid, ursolic acid, and oleanolic acid, citing their MIC values against Candida species from the literature; and (2) an expanded mechanistic discussion that explores possible modes of action based on (a) the structural features of the lupane skeleton, (b) the synergism pattern observed with drugs representing three different antifungal classes (azoles, polyenes, echinocandins), and (c) structural analogues’ known mechanisms. We clearly state that these remain hypotheses requiring direct experimental confirmation.

Minor Comments

COMMENT 1

The manuscript should be carefully revised for English grammar and readability.

RESPONSE

We appreciate this comment. The entire manuscript has been carefully revised for grammar, clarity, and readability. Specific typographical and grammatical errors identified and corrected include: (1) double word "with with" in Section 2.6; (2) "tropicalsis" (a misspelling of "tropicalis") appearing twice in Sections 3.8.1 and 3.8.2; (3) "whiles" replaced with "while" in Section 3.3; (4) a spurious backtick character before "In the study" in the Discussion; and (5) several long, convoluted sentences rephrased for clarity. The manuscript has been reviewed holistically for grammatical accuracy and fluency throughout.

COMMENT 2

Some sentences in the discussion section are too long and could be simplified for better clarity.

RESPONSE

Long compound sentences, particularly in the paragraphs describing biofilm resistance mechanisms, and the study limitations, have been broken into shorter, more readable statements. Scientific content has been preserved throughout these revisions.

COMMENT 3

Scientific names of fungal species should be consistently italicized throughout the manuscript.

RESPONSE

Thank you for pointing this out. We have reviewed the entire manuscript and ensured that all scientific names of fungal species (Candida and specific epithets such as C. albicans, C. glabrata, C. tropicalis, C. krusei, and C. parapsilosis) are consistently italicized throughout.

COMMENT 4

The quality and clarity of some figures and tables should be improved. Figure legends should provide sufficient detail to understand the experiments without referring to the main text.

RESPONSE

All figures have been re-exported at a minimum resolution of 300 DPI suitable for publication. The figure legends for Figs. 2 – 6 have been expanded to be self-contained, now including: a description of what is shown, the number of biological and technical replicates, how data are expressed (mean ± SD), the statistical test applied, and the meaning of all significance symbols and error bars. Table legends have been updated to include complete footnote explanations for all abbreviations used.

COMMENT 5

The conclusions should be slightly moderated since the current results are based only on in vitro experiments.

RESPONSE

The Conclusion section has been revised to clearly reflect the in vitro nature of the present findings and to ensure that statements regarding therapeutic potential are appropriately qualified. Language suggesting direct translational applicability has been replaced with statements emphasizing the need for further investigation, including cytotoxicity assessment and in vivo validation, before therapeutic conclusions can be drawn.

---

## [Decision Letter · Decision Letter 2]

5 May 2026

PONE-D-25-59885R2Potent and Broad-spectrum anti-Candida activity of 6α-(3'-methoxy-4'-hydroxybenzoyl)-lup-20(29)-ene-3-one, a triterpenoid from Paullinia pinnataPLOS One

Dear Dr. Harley,

Thank you for submitting your manuscript to PLOS ONE. After careful consideration, we feel that it has merit but does not fully meet PLOS ONE’s publication criteria as it currently stands. Therefore, we invite you to submit a revised version of the manuscript that addresses the points raised during the review process.

We look forward to receiving your revised manuscript.

Kind regards,

Vartika Srivastava, PhD

Academic Editor

PLOS One

Journal Requirements:

Additional Editor Comments :

Thank you for submitting your manuscript entitled “Potent and Broad-spectrum anti-Candida activity of 6α-(3'-methoxy-4'-hydroxybenzoyl)-lup-20(29)-ene-3-one, a triterpenoid from Paullinia pinnata” to PLOS One. We have now received the reviewers’ comments, which are provided below for your consideration.

Based on the reviewers’ feedback and our editorial assessment, we are pleased to invite you to revise your manuscript. The reviewers have found the study to be of interest and generally well conducted; however, a few minor issues need to be addressed to further improve clarity and presentation.

Reviewers' comments:

Reviewer's Responses to Questions

**Comments to the Author**

1. If the authors have adequately addressed your comments raised in a previous round of review and you feel that this manuscript is now acceptable for publication, you may indicate that here to bypass the “Comments to the Author” section, enter your conflict of interest statement in the “Confidential to Editor” section, and submit your "Accept" recommendation.

Reviewer #2: All comments have been addressed

Reviewer #3: All comments have been addressed

2. Is the manuscript technically sound, and do the data support the conclusions?

Reviewer #2: Yes

Reviewer #3: Yes

3. Has the statistical analysis been performed appropriately and rigorously? 

Reviewer #2: Yes

Reviewer #3: Yes

4. Have the authors made all data underlying the findings in their manuscript fully available?

Reviewer #2: Yes

Reviewer #3: Yes

5. Is the manuscript presented in an intelligible fashion and written in standard English?

Reviewer #2: Yes

Reviewer #3: Yes

6. Review Comments to the Author

Reviewer #2: The manuscript presents a well-conducted and relevant study investigating the antifungal and antibiofilm activity of a triterpenoid isolated from Paullinia pinnata. The revised version has adequately addressed the comments raised in the previous round of review and shows clear improvements in methodological transparency, statistical analysis, and overall presentation.

Only minor editorial points remain:

Slightly reinforce the statement in the Conclusion regarding the absence of cytotoxicity data

Ensure that mechanistic interpretations are consistently presented as hypotheses

Add a brief sentence highlighting the relevance of the antibiofilm findings

Minor language polishing and improved consistency in figure legends

These points are purely editorial and do not affect the scientific validity of the study

Reviewer #3: The graph axes remain very difficult to read due to the pixelated font; However, the available tiffs seem to absolve that issue and make the data interpretable.

7. PLOS authors have the option to publish the peer review history of their article (what does this mean?). If published, this will include your full peer review and any attached files.

Reviewer #2: No

Reviewer #3: No

---

## [Author Response · Author response to Decision Letter 3]

6 May 2026

RESPONSE TO REVIWERS’ COMMRENTS

Journal Requirements:

Comment 1

Response

We thank the journal for this important instruction. We have carried out a thorough review of the entire reference list of the manuscript. Each reference was individually verified against its source to confirm accuracy of author names, article title, journal name, volume, issue, page numbers and year of publication. We confirm that, to the best of our knowledge, none of the references cited in the manuscript have been retracted. No references have been removed or replaced as a result of this check

Reviewer #2:

The manuscript presents a well-conducted and relevant study investigating the antifungal and antibiofilm activity of a triterpenoid isolated from Paullinia pinnata. The revised version has adequately addressed the comments raised in the previous round of review and shows clear improvements in methodological transparency, statistical analysis, and overall presentation.

Only minor editorial points remain:

Comment 1

Slightly reinforce the statement in the Conclusion regarding the absence of cytotoxicity data

Response

The statement on the absence of cytotoxicity data in the Conclusion has been revised. The revised statement more clearly conveys that the absence of mammalian cell toxicity data is a limitation that must be addressed before the therapeutic potential of 6α-(3′-methoxy-4′-hydroxybenzoyl)-lup-20(29)-ene-3-one can be fully assessed.

Comment 2

Ensure that mechanistic interpretations are consistently presented as hypotheses

Response

The mechanism of action section of the Discussion and the antibiofilm discussion paragraph have been reviewed. Sentences where mechanistic claims were not sufficiently framed as hypotheses have been revised.

Comment 3

Add a brief sentence highlighting the relevance of the antibiofilm findings.

A sentence has been added to the antibiofilm discussion paragraph highlighting the therapeutic relevance of the antibiofilm findings. The sentence draws attention to the fact that biofilm formation is a major driver of treatment failure in vulvovaginal candidiasis and that conventional antifungal agents show reduced efficacy against biofilm-associated Candida infections.

Comment 4

Minor language polishing and improved consistency in figure legends

Response

The manuscript has been re-read and minor language corrections made throughout. All figure legends have also been reviewed and revised for consistency in structure, terminology and level of detail across Figs. 3–6.

Fig. 3 and Fig. 4 legends each report MIC data with mean ± SD, state the number of biological and technical replicates, name the statistical test and correction applied, and define all significance symbols. These were already consistent with one another.

Fig. 5 and Fig. 6 legends each report IC50 data, state the number of biological replicates, describe data as mean percentage inhibition ± SD, and state that IC50 values were derived by non-linear regression. These were consistent with one another but differed from Figs. 3 and 4 in the absence of explicit mention of the vehicle control concentration used (1% DMSO). This detail has now been added to both Fig. 5 and Fig. 6 legends for completeness and cross-legend consistency.

Reviewer #3:

Comment 1

The graph axes remain very difficult to read due to the pixelated font; However, the available tiffs seem to absolve that issue and make the data interpretable.

Response

The graph axes have been edited and the writings enlarged to make it easy to read and the data interpretable. We have also the figures using NAAS and ensured that they meet technical requirements.

---

## [Decision Letter · Decision Letter 3]

13 May 2026

Potent and Broad-spectrum anti-Candida activity of 6α-(3'-methoxy-4'-hydroxybenzoyl)-lup-20(29)-ene-3-one, a triterpenoid from Paullinia pinnata

PONE-D-25-59885R3

Dear Dr. Harley,

We’re pleased to inform you that your manuscript has been judged scientifically suitable for publication and will be formally accepted for publication once it meets all outstanding technical requirements.

Kind regards,

Vartika Srivastava, PhD

Academic Editor

PLOS One

Additional Editor Comments (optional):

Reviewers' comments:

Reviewer's Responses to Questions

**Comments to the Author**

1. If the authors have adequately addressed your comments raised in a previous round of review and you feel that this manuscript is now acceptable for publication, you may indicate that here to bypass the “Comments to the Author” section, enter your conflict of interest statement in the “Confidential to Editor” section, and submit your "Accept" recommendation.

Reviewer #2: All comments have been addressed

2. Is the manuscript technically sound, and do the data support the conclusions?

Reviewer #2: Yes

3. Has the statistical analysis been performed appropriately and rigorously? 

Reviewer #2: Yes

4. Have the authors made all data underlying the findings in their manuscript fully available?

Reviewer #2: (No Response)

5. Is the manuscript presented in an intelligible fashion and written in standard English?

Reviewer #2: Yes

6. Review Comments to the Author

Reviewer #2: The manuscript presents a scientifically sound and well-conducted study investigating the anti-Candida and antibiofilm activities of a triterpenoid isolated from Paullinia pinnata. The experimental design is appropriate, the methodologies are adequately described, and the data generally support the conclusions presented by the authors.

The revised version has substantially improved following the previous rounds of review. The authors have adequately addressed the concerns raised regarding methodological transparency, statistical analysis, figure presentation, and interpretation of mechanistic aspects. The inclusion of antifungal combination studies and antibiofilm assays further strengthens the relevance of the work, particularly in the context of increasing antifungal resistance among Candida species.

The manuscript is clearly written, the results are presented in a coherent manner, and the conclusions are appropriately moderated, especially regarding the need for future cytotoxicity studies before therapeutic applications can be considered.

Overall, I believe the manuscript is suitable for publication in PLOS ONE in its current form.

7. PLOS authors have the option to publish the peer review history of their article (what does this mean?). If published, this will include your full peer review and any attached files.

Reviewer #2: No

---

## [Editor Report · Acceptance letter]

PONE-D-25-59885R3

PLOS One

Dear Dr. Harley,

I'm pleased to inform you that your manuscript has been deemed suitable for publication in PLOS One. Congratulations! Your manuscript is now being handed over to our production team.

Kind regards,

on behalf of

Dr. Vartika Srivastava

Academic Editor

PLOS One